# TREESYNTH: Synthesizing Diverse Data from Scratch via Tree-Guided Subspace Partitioning

**Sheng Wang**[*], **Pengan Chen**[*], **Jingqi Zhou**[*], **Qintong Li**, **Jingwei Dong**
The University of Hong Kong
{u3009618, cpa2001, u3011211, qtli, joviedong89}@connect.hku.hk

**Jiahui Gao**
The University of Hong Kong
ggaojiahui@gmail.com

**Boyang Xue, Jiyue Jiang**
The Chinese University of Hong Kong
byxue@se.cuhk.edu.hk, jiangjy@link.cuhk.edu.hk

**Lingpeng Kong, Chuan Wu**
The University of Hong Kong
{lpk, cwu}@cs.hku.hk

## Abstract

Model customization necessitates high-quality and diverse datasets, but acquiring such data remains time-consuming and labor-intensive. Despite the great potential of large language models (LLMs) for data synthesis, current approaches are constrained by limited seed data, model biases, and low-variation prompts, resulting in limited diversity and biased distributions with the increase of data scales. To tackle this challenge, we introduce TREESYNTH, a tree-guided subspace-based data synthesis approach inspired by decision trees. It constructs a spatial partitioning tree to recursively divide a task-specific full data space (*i.e.*, root node) into numerous atomic subspaces (*i.e.*, leaf nodes) with mutually exclusive and exhaustive attributes to ensure both distinctiveness and comprehensiveness before synthesizing samples within each atomic subspace. This globally dividing-and-synthesizing method finally collects subspace samples into a comprehensive dataset, effectively circumventing repetition and space collapse to ensure the diversity of large-scale data synthesis. Furthermore, the spatial partitioning tree enables sample allocation into atomic subspaces, allowing the rebalancing of existing datasets for more balanced and comprehensive distributions. Empirically, extensive experiments across diverse benchmarks consistently demonstrate the superior data diversity, model performance, and robust scalability of TREESYNTH compared to both human-crafted datasets and peer data synthesis methods, with an average performance gain reaching 10%. Besides, the consistent improvements of TREESYNTH-balanced datasets highlight its efficacious application to redistribute existing datasets for more comprehensive coverage and the induced performance enhancement. The code is available at https://github.com/cpa2001/TreeSynth.

## 1 Introduction

With the superior performance, large language models (LLMs), such as OpenAI o1 [1], LLaMA-3 [2], and DeepSeek R1 [3], have been deployed for various downstream applications, including code copilot [4], mathematical reasoning [5], psychology [6], etc. The success of these models

---

[*]Equal Contribution.

39th Conference on Neural Information Processing Systems (NeurIPS 2025).

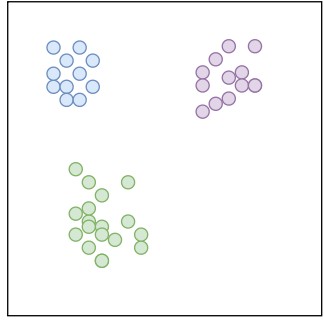 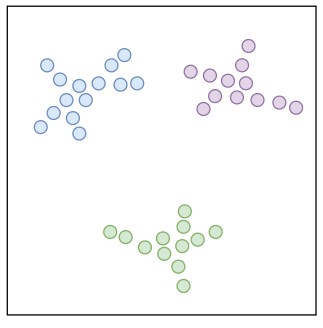 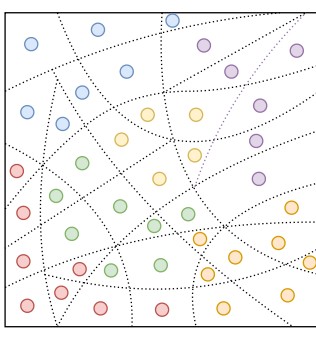

| (a) Temperature Sampling | (b) Evol-Instruct | (c) TREESYNTH |

Figure 1: Intuitive comparison of Temperature Sampling, Evol-Instruct, and TREESYNTH. Temperature Sampling typically generates a specific data distribution induced by model biases, while Evol-Instruct evolves seed data along specified directions. In contrast, TREESYNTH starts from a global perspective by dividing the entire data space into mutually exclusive and complementary subspaces before sampling from each subspace, resulting in a more balanced and diverse dataset with comprehensive coverage.

largely depends on the availability of large-scale diverse training datasets. However, open-access data are typically drying up [7, 8], and manual data curation is both time-consuming and labor-intensive [9, 10], hindering its availability. This necessitates a novel approach to continuously generate data that supports the ongoing advancement of LLMs across different domains.

To customize LLMs and further enhance their specific capabilities, synthesizing domain-specific data using their remarkable abilities emerges as a promising solution [11, 12]. Pioneering approaches typically paraphrase current datasets [9, 13], or prompt existing LLMs to reproduce their training data [14, 15]. However, due to the inherent model biases and minimal-variation prompts, the generated data often suffers repetition and homogeneity. To remedy this, increasing sampling temperature [16] increases data diversity yet reduces quality. In contrast, attribute-driven approaches (*e.g.*, Persona Hub [17]) utilize the in-context learning capabilities of LLMs to offer increased diversity and improved quality simultanously. Alternatively, evolving new data from existing datasets, represented by Evol-Instruct [18]), augment existing data along different directions to induce diverse generation. More details about the related works are provided in Appendix. 5. However, as shown in Figure 1, from the perspective of data space, these methods typically start from the local distribution (*i.e.*, model biases, seed data, or low-variation prompts) without the global view, hindering their comprehensive coverage. This raises the following question:

*"Is there an automatic solution that starts from a global perspective to fully cover the domain-specific data space for higher diversity?"*

To achieve this objective, we introduce TREESYNTH, a tree-guided subspace-based data synthesis approach inspired by decision trees [19]. It consists of two key stages: data space partitioning and subspace data synthesis. During the former phase, as illustrated in Figure 2, TREESYNTH employs a spatial partitioning tree to recursively divide a task-specific whole data space (*i.e.*, root node defined by textual descriptions) into numerous atomic subspaces (*i.e.*, leaf nodes). These subspaces are characterized by mutually exclusive and exhaustive attribute values to ensure both distinctiveness and diversity. In the subsequent subspace data synthesis phase, samples are generated within each subspace separately, before collecting them as a diverse and comprehensive dataset. By employing this globally divide-and-synthesize methodology, TREESYNTH effectively prevents repetition and space collapse to ensure the diversity and completeness of large-scale data synthesis, successfully avoiding the drawbacks of previous methods. Additionally, the spatial partitioning tree enables the allocation of samples into atomic subspaces, thereby allowing the re-balancing of existing datasets for more balanced and comprehensive distributions. Extensive experiments with both open-source and closed-source models across diverse benchmarks, spanning mathematical reasoning, code generation and psychology, demonstrate that TREESYNTH consistently achieves the best downstream performance with superior data diversity compared to both human-crafted datasets and peer data synthesis methods, with the average performance enhancement reaching 10%, underscoring its great effectiveness and generalization. Besides, the linear (or even better) performance growth trajectories with increased data volume highlight TREESYNTH's remarkable robustness and scalability for large-

scale data synthesis. Furthermore, the improved results achieved by applying TREESYNTH to the synthetic datasets demonstrate its efficacious application to redistribute existing datasets for more comprehensive coverage and the induced performance enhancement.

The main contributions are summarized as follows:

- We propose TREESYNTH, a tree-guided subspace-based data synthesis approach, which features mutually exclusive and exhaustive subspace partitioning, effectively circumventing repetition and space collapse to ensure the diversity of large-scale data synthesis.

- Extensive experiments consistently highlight TREESYNTH's superior data diversity, model performance and robust scalability over human-crafted datasets and peer synthesis methods.

- The sample allocation of TREESYNTH allows re-balancing existing datasets for more comprehensive coverage, leading to empirically verified downstream performance enhancement.

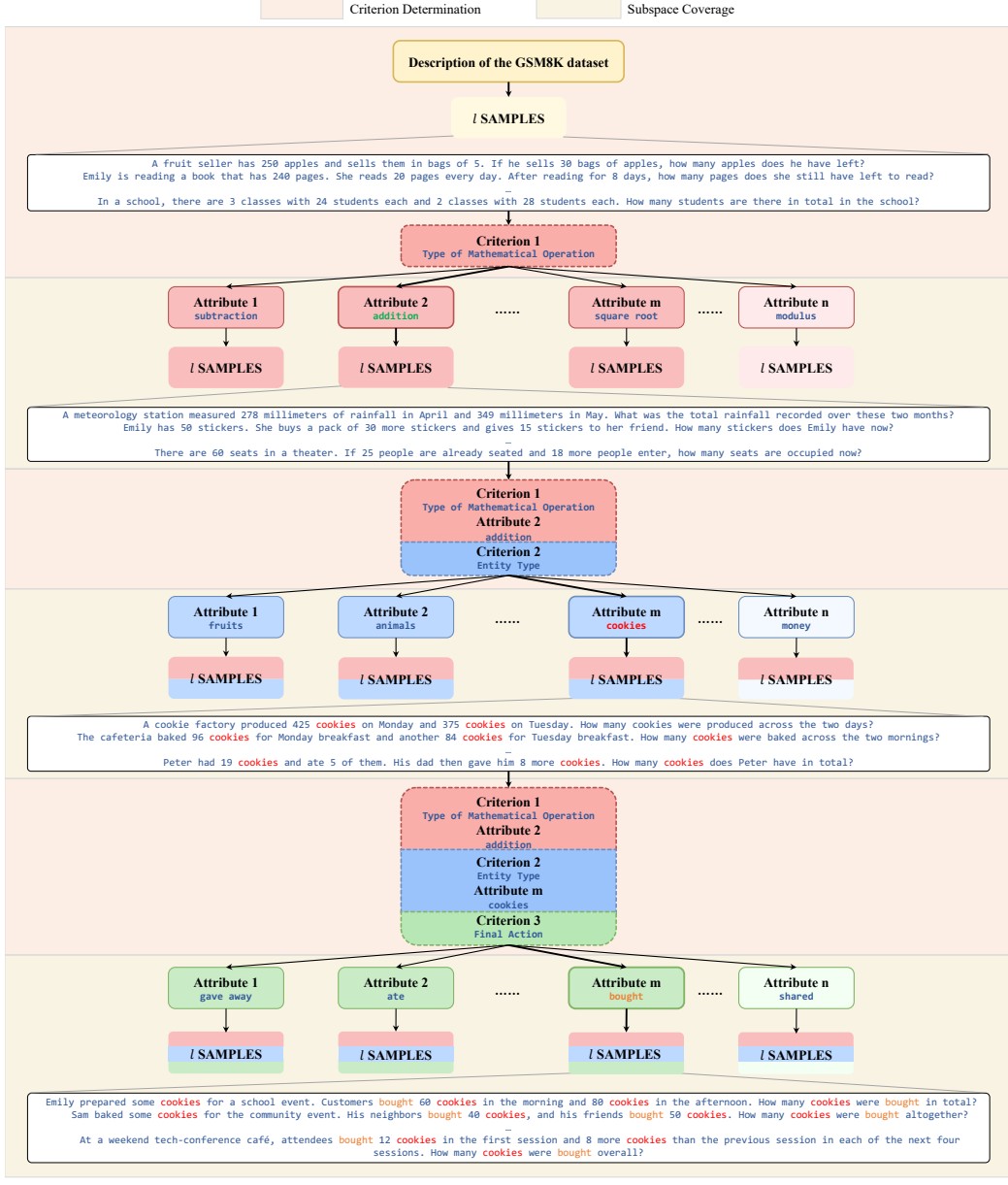

Figure 2: A spatial partitioning tree visualization of TREESYNTH, exemplified through GSM8K-style data synthesis.

## 2 Preliminary Knowledge

As a canonical machine learning algorithm, decision trees [19] are widely recognized for their simplicity, efficiency, and strong interpretability. For any given sample, the decision tree recursively allocates it to deeper nodes within the hierarchical structure, until it reaches one and only one leaf node. This functionality relies on two essential characteristics: (1) All leaf nodes of any sub-tree starting from the root node are mutually complementary, ensuring every sample can be allocated to at least one leaf node. (2) All leaf nodes of such a sub-tree maintain mutual exclusivity, guaranteeing each sample can be assigned to at most one leaf node.

From a spatial perspective, the root node represents the entire sample space. As the tree delves deeper with each layer of nodes, the space is exhaustively and exclusively divided into multiple subspaces (*i.e.*, child nodes). Hence, for any given task, we can conceptualize its training data as the entire space (*i.e.*, root node), allowing to establish a mapping between the nodes of decision tree and the training data subspaces. In detail, the decision tree partitions the entire training data space into multiple leaf nodes, with each leaf node corresponding to a data subspace with specific attributes.

This inspires us to leverage the subspace partitioning of decision trees for data synthesis, offering two notable advantages: (1) **Diversity**: The exclusivity of leaf nodes ensures the variation across different subspaces, thereby guaranteeing samples diversity. (2) **Comprehensive Coverage**: The complementarity and exhaustiveness of leaf nodes ensures the sampling of comprehensive data, preventing sample collapse.

## 3 Method

Inspired by the mapping between a decision tree and data space, we propose TREESYNTH, a tree-guided subspace-based data synthesis approach. It consists of two primary stages: data space partitioning and subspace data synthesis. The first stage generates a spatial partitioning tree $\mathcal{T}$, analogous to the construction of a decision tree, while the second synthesizes data within each atomic subspaces (*i.e.*, leaf node). Beyond data synthesis, we also elaborate how TREESYNTH can re-balance existing datasets, facilitating more comprehensive coverage and induced performance improvement.

### 3.1 Data Space Partitioning

Given any data space $\mathcal{S}$ (*i.e.*, any node in the tree) with its context description $\mathcal{C}_\mathcal{S}$, data space partitioning aims to decompose it into multiple subspaces $\mathcal{S}_{\text{sub}} = \{s_i | i = 1, 2, ..., n\}$. As shown in Figure 3a, the partitioning process mirrors the construction of decision trees, and comprises two critical steps: Criterion Determination and Subspace Coverage. These steps ensure the mutual exclusivity (*i.e.*, $\forall p \neq q, s_p \cap s_q = \varnothing$) and exhaustiveness (*i.e.*, $\bigcup_{i=1}^{n} s_i = \mathcal{S}$) among subspaces, respectively. This suggests that each subspace is disjoint, and collectively, they fully encompass the original space.

**Criterion Determination.** The essence of this step lies in selecting a criterion $\delta$ that most effectively differentiates data within the space $\mathcal{S}$ so that most data characteristics can be captured with minimal criteria. Specifically, according to the space description $\mathcal{C}_\mathcal{S}$ (*e.g.*, "GSM8K-style mathematical questions" as shown in Figure 2), an LLM is firstly deployed to generate $l$ maximally diverse pivot samples $\mathcal{X} = \{x_t | t = 1, 2, ..., l\}$ to approximate the whole space $\mathcal{S}$. Subsequently, another off-the-shelf LLM, instructed to proficiently identify inter-sample distinctions, determines exactly one core criterion $\delta$ (*e.g.*, Type of Mathematical Operation). This criterion $\delta$ optimally partitions $\mathcal{X}$ into mutually exclusive attribute values $V_\mathcal{X}^\delta = \{v_j^\delta | j = 1, 2, ..., m\}$ (*e.g.*, addition and subtraction), categorizing each sample $x_t \in \mathcal{X}$ into exactly one attribute value to ensure mutual exclusivity across child nodes.

**Subspace Coverage.** Despite the existing attribute values $V_\mathcal{X}^\delta$, the mutually exclusive subspace $\mathcal{S}_\mathcal{X}^\delta$, derived from partitioning $\mathcal{X}$ with these values, may not exhaustively cover the original space $\mathcal{S}$ due to a limited number $l$ of pivot samples. This imposes the risk of non-complementarity among the child nodes. Hence, subspace coverage is designed to supplement potential attribute values of criterion $\delta$ to comprehensively model the entire data space $\mathcal{S}$. Specifically, we instruct an LLM to expand the attribute values $V_\mathcal{X}^\delta$ to $V_\mathcal{S}^\delta = \{v_i^\delta | i = 1, 2, ..., m, m+1, ..., n\}$ (*e.g.*, additionally including square root and modulus). The expanded attribute values $V_\mathcal{S}^\delta$ must be non-overlapping and fully cover the criterion $\delta$. Consequently, the exhaustive and exclusive subspaces $\mathcal{S}_{\text{sub}} = \{s_i | i = 1, 2, ..., m, m+1, ..., n\}$ can be generated by $s_i = \mathcal{C}_\mathcal{S} \cap v_i^\delta$ for each $i$, completely filling the data space $\mathcal{S}$.

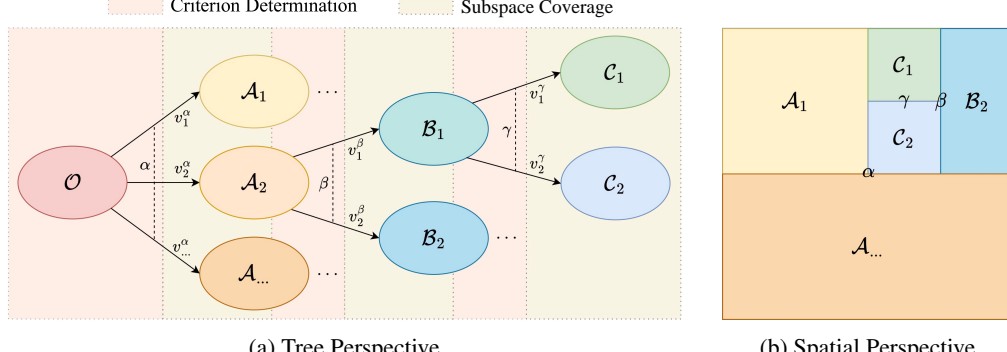

| (a) Tree Perspective | (b) Spatial Perspective |

Figure 3: Illustration of TREESYNTH. (a) Data space partitioning iterates criterion determination and subspace coverage. The former identifies the criteria (*e.g.*, $\alpha$, $\beta$, $\gamma$) and their associated attribute values (*e.g.*, $v_1^\alpha$, $v_2^\alpha$, $v_1^\beta$, $v_2^\beta$, $v_1^\gamma$, $v_2^\gamma$) to divide current nodes (*e.g.*, entire space $\mathcal{O}$, $\mathcal{A}_1$, $\mathcal{A}_2$, $\mathcal{A}_{...}$, $\mathcal{B}_1$, $\mathcal{B}_2$) until reaching the leaf nodes (*e.g.*, atomic subspaces $\mathcal{C}_1$, $\mathcal{C}_2$, ...), while the latter complements potential attribute values (*e.g.*, $v_{...}^\alpha$) to ensure exhaustive coverage of the entire data space. (b) The spatial visualization depicts the mapping between tree nodes and data subspaces, highlighting the mutually exclusiveness and exhaustiveness of the subspaces.

However, not all criteria can be exhaustively enumerated. For example, if a criterion standard is numerical values of mathematical questions, it contains infinite attribute values (*e.g.*, 0, 1, 2, 3 ...). In such cases, we set a maximum number of attribute values $N$. Once the number of attribute values $n$ exceeds $N$, we refrain from setting individual sub-nodes for each attribute value, and instead establish an infinite node encompassing potential attribute values. Whenever needed, one attribute value is randomly sampled from all potential candidates. This effectively prevents the generation of numerous trivial child nodes, thereby reducing the redundancy of the tree.

**Spatial Partitioning Workflow.** Recursively, we apply both criterion determination and subspace coverage steps to construct a complete spatial partitioning tree. As illustrated in Figure 2 and 3, starting from the entire data space $\mathcal{O}$ (*i.e.*, root node $\mathcal{N}_{Root}$) represented by training data description $\mathcal{C}_\mathcal{O}$, we first perform criterion determination on $\mathcal{O}$ to identify the optimal criterion $\alpha$ that most effectively distinguishes data within the space $\mathcal{O}$. Through subsequent subspace coverage, $\mathcal{O}$ is partitioned into mutually exclusive and exhaustive subspaces $\mathcal{O}_{\text{sub}} = \{\mathcal{A}_k | k = 1, 2, ...\}$ based on $\alpha$. Subsequently, the breadth-first search (BFS) algorithm is applied to each subspace $\mathcal{A}_k$ to recursively execute both steps until reaching the maximal depth $d$. As shown in Figure 3a, $\mathcal{A}_2$ is further divided into $\mathcal{B}_1$ and $\mathcal{B}_2$, with $\mathcal{B}_1$ subsequently partitioned into the leaf nodes $\mathcal{C}_1$ and $\mathcal{C}_2$. Finally, a spatial partitioning tree $\mathcal{T}$ is constructed and decomposes $\mathcal{O}$ into numerous mutually exclusive and complementary atomic subspaces $\mathcal{O}_{\text{Leaf}}^*$, each corresponding to a leaf node $\mathcal{N}_{\text{Leaf}}^*$. We also present the pseudo code to formularize the whole process in Algorithm 1. **The mutual exclusivity of leaf nodes intrinsically ensures diversity in the synthesized dataset, while their exhaustiveness guarantees comprehensive coverage of the data space.** The dual properties effectively prevents data collapse observed in previous data synthesis methods.

## 3.2 Subspace Data Synthesis

The objective of subspace data synthesis stage is to create data within mutually exclusive and complementary atomic subspaces $\mathcal{O}_{\text{Leaf}}^*$ defined by spatial partitioning tree $\mathcal{T}$, ultimately producing a diverse and balanced dataset with comprehensive space coverage. Specifically, for each leaf node $\mathcal{N}_{\text{Leaf}}^*$, we first compile its description along the hierarchical path from the root node $\mathcal{N}_{Root}$ to itself. This path can be formally expressed as $\mathcal{N}_{Root} \rightarrow v_i^\alpha \rightarrow v_j^\beta \rightarrow \cdots \rightarrow v_k^\gamma \rightarrow \mathcal{N}_{\text{Leaf}}^*$, where $\{v_i^\alpha, v_j^\beta, \cdots, v_k^\gamma\}$ denotes the individual attribute values of parent nodes along the path. Similar to the generation of pivot samples, we combine both $\mathcal{C}_\mathcal{O}$ and the attribute value sequence $\{v_i^\alpha, v_j^\beta, \cdots, v_k^\gamma\}$ as the description of $\mathcal{O}_{\text{Leaf}}^*$, and instruct an LLM to generate $N_{\text{Leaf}}$ samples distributed within its subspace. As depicted in Figure 1c, by collecting data generated within all the leaf nodes, we obtain a final dataset with high diversity, balanced distribution, and comprehensive coverage.

## 3.3 TREESYNTH-Guided Data Balance

Beyond data synthesis, TREESYNTH can also be leveraged to optimize existing datasets for improved balance and comprehensiveness. Given that TREESYNTH synthesizes data from scratch, a spatial

partitioning tree $\mathcal{T}_\mathcal{D}$ can be constructed solely based on the context description $\mathcal{C}_\mathcal{D}$ (*i.e.*, full space $\mathcal{O}_\mathcal{D}$) of a given dataset $\mathcal{D} = \{d_u | u = 1, 2, ..., w\}$. Thanks to the mutually exclusive and exhaustive partitioning, every sample $d_u$ can be systematically routed through successive levels of the hierarchy, ultimately landing in a unique leaf node (*i.e.*, atomic subspace). The distribution of all the samples across leaf nodes reveals the dataset's coverage pattern within the full space. To regulate the distribution across subspaces, a threshold $N_{\text{Sub}}$ is introduced. Subspaces containing more than $N_{\text{Sub}}$ samples are randomly downsampled to reduce overrepresentation, while those with fewer samples are augmented with TREESYNTH to meet the threshold. The integration of all adjusted samples yields a new dataset $\mathcal{D}_{\text{balance}}$ with more comprehensive coverage and better balance than the vanilla one.

## 4 Experiments

### 4.1 General Setup

**Benchmarks.** To comprehensively evaluate the advantages of TREESYNTH, we compare it against several baselines across diverse benchmarks. For data synthesis, we first apply standard mathematical reasoning and code generation tasks, including GSM8K [20], MATH [21], MBPP [22] and HumanEval [23], to assess TREESYNTH's data diversity, model performance improvement, and scalability. Besides, we employ SimpleToM [24], a psychological task, to further examine TREESYNTH's effectiveness in promoting data balance. More details are elaborated in Section A.2.

**Base Models.** For generation models to synthesize data with different methods, we employ both open-source (*i.e.*, `LLaMA3.3-70B-Instruct` [2] and `Qwen2.5-72B-Instruct` [25]) and closed-source (*i.e.*, `GPT-4o`[2]) models. To compare the performance of TREESYNTH and baselines, we fine-tune two popular open-source foundation LLMs (*i.e.*, `LLaMA3.1-8B` [2] and `Qwen2.5-7B` [25]) on the perspective generated data. These models are chosen for their leading performance and popularity.

**Baselines.** With standard Zero-Shot and Few-Shot performance as reference, we evaluate the effectiveness of TREESYNTH by comparing it with two categories of baselines. The first category comprises human-curated datasets (*i.e.*, Vanilla Data): the training sets of GSM8K (7,473 samples) and MATH (7,500 samples) for mathematical reasoning, and Code Alpaca [26] (2,689 samples[3]) for code generation. The second category consists of LLM-synthesized training data, covering three representative methods: Temperature Sampling [16], seed-driven method (*i.e.*, Evol-Instruct [18]), and attribute-driven method (*i.e.*, Persona Hub [17]). Each method synthesizes 100k samples in the styles of GSM8K, MATH, and Code Alpaca, and 40k samples in SimpleToM style. Further details on the baselines and implementation are provided in Appendix A.3 and A.4, respectively.

### 4.2 Main Results

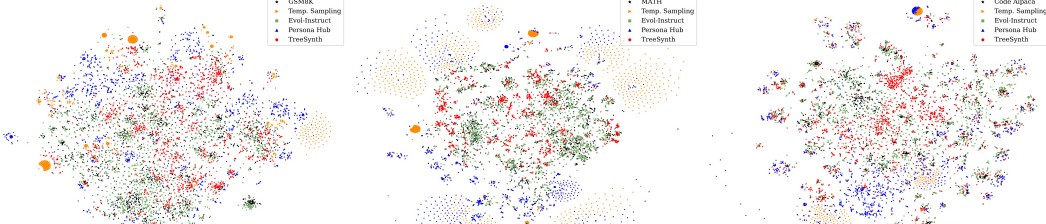

| (a) GSM8K | (b) MATH | (c) Code Alpaca |

Figure 4: t-SNE visualization of `LLaMA3.3-70B-Instruct`-synthesized datasets for various methods across GSM8K, MATH, and Code Alpaca styles.

**TREESYNTH exhibits substantially better data diversity and more comprehensive coverage across various tasks and models than both human-curated datasets and peer synthetic methods.** As shown in Tables 1, 2, and 4, we compare the data diversity of TREESYNTH and peer methods driven by `Qwen2.5-72B-Instruct`, `LLaMA3.3-70B -Instruct`, and `GPT-4o`, respectively. Temperature Sampling consistently yields lower diversity than vanilla training datasets. Although Evol-Instruct and Persona Hub show improvements over vanilla ones in some cases, they generally deteriorate on MATH benchmark, suggesting their limited robustness. Compared to all the baselines, TREESYNTH

---

[2]`https://openai.com/index/hello-gpt-4o/`
[3]Only Python-related samples are retained, aligning with HumanEval and MBPP.

| Method | GSM8K↑ | Diversity↓ | MATH↑ | Diversity↓ | MBPP↑ | HumanEval↑ | Diversity↓ | Avg.↑ |
|---|---|---|---|---|---|---|---|---|
| *Foundation Model:* LLAMA-3.1 8B | | | | | | | | |
| Zero-Shot | 4.85 | - | 3.54 | - | 19.8 | 15.85 | - | 11.01 |
| Few-Shot | 40.26 | - | 20.46 | - | - | - | - | - |
| Vanilla Data | 58.15 | 0.40 | 19.48 | 0.16 | 46.8 | 43.29 | 0.29 | 41.93 |
| Temp. Sampling | 55.42 | 0.44 | 22.08 | 0.37 | 44.6 | 41.46 | 0.33 | 40.89 |
| Evol-Instruct | 63.46 | 0.37 | 27.26 | **0.15** | 40.6 | 41.46 | **0.22** | 43.20 |
| Persona Hub | 61.41 | **0.35** | 23.78 | 0.34 | 45.8 | 40.24 | 0.26 | 42.81 |
| TREESYNTH | **69.45** | 0.36 | **27.52** | **0.15** | **50.2** | **48.17** | 0.23 | **48.84** |
| *Foundation Model:* QWEN-2.5 7B | | | | | | | | |
| Zero-Shot | 54.97 | - | 54.38 | - | 11.2 | 54.88 | - | 43.86 |
| Few-Shot | 67.40 | - | 47.58 | - | - | - | - | - |
| Vanilla Data | 68.76 | 0.40 | 47.68 | 0.16 | 53.4 | 77.44 | 0.29 | 61.82 |
| Temp. Sampling | 69.67 | 0.41 | 61.70 | 0.37 | 54.8 | 76.83 | 0.33 | 65.75 |
| Evol-Instruct | 75.13 | 0.37 | 59.60 | **0.15** | 55.8 | 76.83 | **0.22** | 66.84 |
| Persona Hub | 82.79 | **0.35** | 61.98 | 0.34 | 58.8 | 75.00 | 0.26 | 69.64 |
| TREESYNTH | **85.44** | 0.36 | **63.28** | **0.15** | **59.4** | **78.05** | 0.23 | **71.54** |

Table 1: Model performance and data diversity comparison of various methods with `LLaMA3.3-70B-Instruct`-powered data synthesis across two foundation models and multiple benchmarks. "Zero-Shot" and "Few-Shot" exhibit the base performance of foundation models. "Temp. Sampling" is abbreviated from "Temperature Sampling". "Vanilla Data" denotes the original GSM8K and MATH training sets, and the Code Alpaca Python subset for HumanEval and MBPP. "Diversity" is measured by cosine similarity, where lower values indicate greater diversity. Bold and underlining indicate the best and second-best indicators, respectively. "Avg." means the average of the performance scores across all the benchmarks.

consistently achieves the best diversity across almost all the benchmarks and generation models. Notably, the `GPT-4o`-powered TREESYNTH exhibits diversity enhancements of 12.5%, 25.0%, and 34.5% over the vanilla GSM8K, MATH, and Code Alpaca datasets, respectively. In addition, we generate sentence embeddings[4] of instructions synthesized by various methods powered by `LLaMA3.3-70B-Instruct`, and visualize their distributions via t-SNE method [27] in Figure 4. Apparently, Temperature Sampling and Persona Hub exhibit concentrated distributions in limited subspaces, revealing severe diversity constraints stemming from inherent biases. Evol-Instruct's distributions mirror those of vanilla datasets, demonstrating strong dependence on source data. In contrast, TREESYNTH, strategically partitioning the full space from a global perspective and synthesizing data within subspaces, eliminates inherent model biases and transcends source data limitations. Briefly, both the observation on diversity indicators and visualization confirm the efficacy of TREESYNTH in producing diverse and comprehensive datasets across various domains and models, effectively alleviating inherent model biases and constraints imposed by initial datasets.

**Models trained on TREESYNTH data consistently outperform those trained on both human-crafted datasets and synthetic baselines across all the tasks, foundation and generation models.** Specifically, for a fair comparison of different methods, we train models using randomly sampled subsets from each synthetic dataset, matching the sizes of the corresponding vanilla training sets. As shown in Tables 1, 2 and 4, all synthetic methods—Temperature Sampling, Evol-Instruct, and Persona Hub—generally surpass human-curated datasets on average, highlighting the limitations of manual data construction. Thanks to the seed-driven and attribute-driven design, Evol-Instruct and Persona Hub further outperform Temperature Sampling, aligning with the claims in their original works [18, 17]. More microscopically, these methods, however, do not yield stable improvements across all benchmarks. For instance, in Table 1, their performance on the HumanEval benchmark falls below that of the vanilla dataset, indicating limited robustness. In contrast, TREESYNTH delivers the best performance across all tasks, foundation and generation models than all baselines without exception. Notably, training `Qwen2.5-7B` on TREESYNTH data synthesized by `GPT-4o` yields an average performance improvement of over 10% compared to the original training sets, underscoring the effectiveness and robust generalization capabilities of TREESYNTH.

**With the global data spatial perspective guided by tree structure, TREESYNTH effectively scales datasets while preserving data quality, suggesting great scalability wherein downstream performance consistently improves with increased data volume.** As shown in Figure 6, 5 and 8,

---

[4]We utilize the popular all-mpnet-base-v2 model, available at `https://huggingface.co/sentence-transformers/all-mpnet-base-v2`.

| Method | GSM8K↑ | Diversity↓ | MATH↑ | Diversity↓ | MBPP↑ | HumanEval↑ | Diversity↓ | Avg.↑ |
|---|---|---|---|---|---|---|---|---|
| | | | *Foundation Model:* LLAMA-3.1 8B | | | | | |
| Zero-Shot | 4.85 | - | 3.54 | - | 19.8 | 15.85 | - | 11.01 |
| Few-Shot | 40.26 | - | 20.46 | - | - | - | - | - |
| Vanilla Data | 58.15 | 0.40 | 19.48 | 0.16 | 46.8 | 43.29 | 0.29 | 41.93 |
| Temp. Sampling | 54.97 | 0.45 | 24.28 | 0.29 | 44.8 | 45.73 | 0.32 | 42.45 |
| Evol-Instruct | 61.03 | 0.39 | 24.58 | 0.19 | 45.2 | 49.39 | 0.25 | 45.05 |
| Persona Hub | 63.38 | **0.35** | 27.74 | 0.28 | 45.2 | 45.12 | 0.29 | 45.36 |
| TREESYNTH | **66.72** | 0.35 | **30.34** | **0.12** | **50.8** | **50.00** | 0.19 | **49.46** |
| | | | *Foundation Model:* QWEN-2.5 7B | | | | | |
| Zero-Shot | 54.97 | - | 54.38 | - | 11.2 | 54.88 | - | 43.86 |
| Few-Shot | 67.40 | - | 47.58 | - | - | - | - | - |
| Vanilla Data | 68.76 | 0.40 | 47.68 | 0.16 | 53.4 | 77.44 | 0.29 | 61.82 |
| Temp. Sampling | 80.67 | 0.45 | 62.76 | 0.29 | 59.6 | **80.49** | 0.32 | 70.88 |
| Evol-Instruct | 73.16 | 0.39 | 61.10 | 0.19 | 59.2 | **80.49** | 0.25 | 68.49 |
| Persona Hub | 83.24 | **0.35** | 66.22 | 0.28 | 61.6 | 77.44 | 0.29 | 72.12 |
| TREESYNTH | **86.13** | 0.35 | **66.84** | **0.12** | **62.8** | **80.49** | 0.19 | **74.06** |

Table 2: Model performance and data diversity comparison of various methods with `GPT-4o`-powered data synthesis across two foundation models and multiple benchmarks. "Zero-Shot" and "Few-Shot" exhibit the base performance of foundation models. "Temp. Sampling" is abbreviated from "Temperature Sampling". "Vanilla Data" denotes the original GSM8K and MATH training sets, and the Code Alpaca Python subset for HumanEval and MBPP. "Diversity" is measured by cosine similarity, where lower values indicate greater diversity. Bold and underlining indicate the best and second-best indicators, respectively. "Avg." means the average of the performance scores across all the benchmarks.

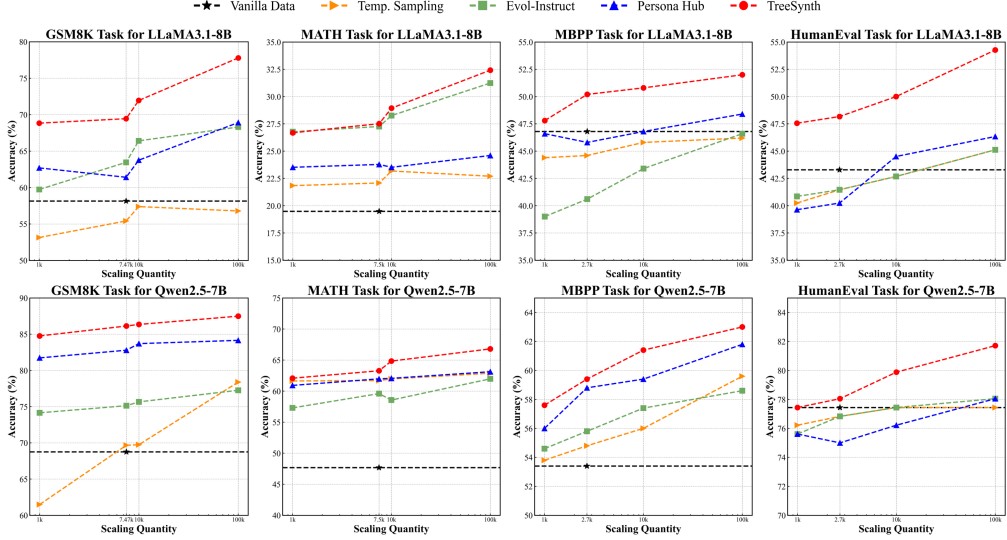

Figure 5: Model performance trends across data scales for different methods powered by `LLaMA3.3-70B-Instruct`. "Temp. Sampling" refers to Temperature Sampling. "Vanilla Data" denotes original GSM8K and MATH training sets, and Code Alpaca Python subset for HumanEval and MBPP.

we evaluate model performance across synthetic datasets of 1k, 10k and 100k samples, as well as at a scale equivalent to the corresponding vanilla training sets. Human-curated datasets inherently suffer from limited scalability due to the prohibitive cost of manual annotation. Despite exhibiting linear growth trends occasionally, all the baselines (*i.e.*, Temperature Sampling, Persona Hub, and Evol-Instruct) encounter performance saturation with diminishing improvements as dataset volume increases in nearly half of the evaluated settings, and even suffer from degradation in some cases, reflecting their instability. As mentioned above, this can be attributed to the intensification of low-variation prompts, model and seed data biases. In contrast, TREESYNTH not only remarkably surpasses all the baselines on all data scales, but also exhibits approximately linear (even better) performance growth with increasing data volume, underscoring its superior scalability. Besides, with the globally spatial perspective circumventing the local distribution biases, TREESYNTH still exhibits

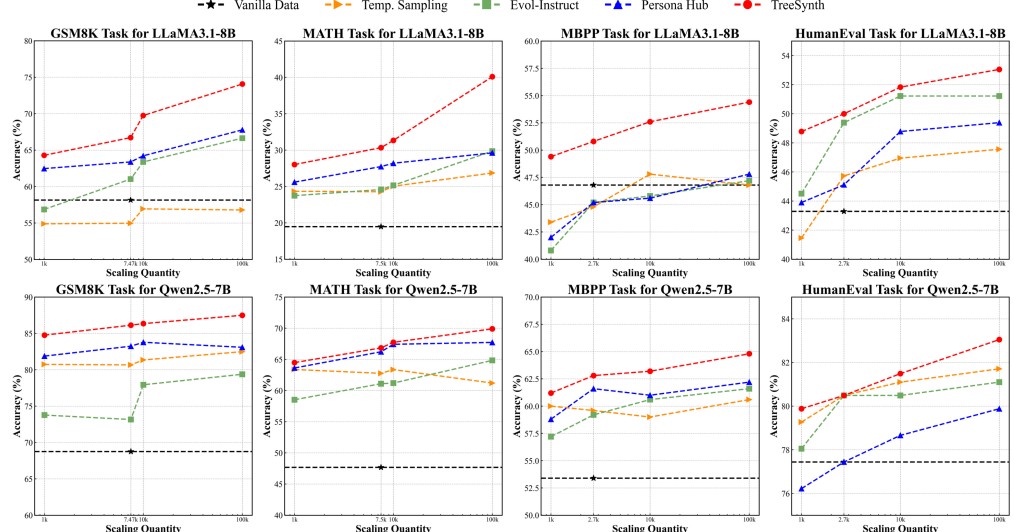

Figure 6: Model performance trends across data scales for different methods powered by `GPT-4o`. "Temp. Sampling" refers to Temperature Sampling, while "Vanilla Data" denotes original GSM8K and MATH training sets, and Code Alpaca Python subset for HumanEval and MBPP.

| Performance | Temp. Sampling | Persona Hub | TREESYNTH | TREESYNTH-balanced | |
| --- | --- | --- | --- | --- | --- |
| | | | | Temp. Sampling | Persona Hub |
| **Accuracy** | 78.9 | 85.7 | 88.0 | 86.8 | **88.6** |
| **Diversity** | 0.37 | 0.33 | 0.33 | 0.33 | **0.32** |

Table 3: The diversity of data generated by different methods powered by `LLaMA3.3-70B-Instruct` and its TREESYNTH-Guided Data Balanced version, along with the performance of `LLaMA3.1-8B` trained on these datasets on the SimpleToM benchmark. "Div." stands for Diversity, a metric assessed by computing cosine similarity among data points within the dataset, where lower numerical values directly indicate higher diversity levels.

steeper performance growth trajectories beyond the 10k sample scale, indicating its great potential for large-scale data synthesis. Detailed numerical results are also provided in Table 5, 6, 7 and 8 in Appendix A.5 for more precise reference.

## 4.3 TREESYNTH-Guided Data Balance

**Beyond data synthesis, TREESYNTH significantly enhances the distributional balance of existing datasets, effectively improving data diversity and downstream model performance.** Specifically, we repeat the `LLaMA3.3-70B-Instruct`-powered experiments on the `LLaMA3.1-8B` foundation model in Section 4.2 on the SimpleToM benchmark, but additionally apply the TREESYNTH-guided data balance technique to the synthetic datasets of Temperature Sampling and Persona Hub[5]. As listed in Table 3, TREESYNTH continues to outperform other unbalanced approaches. Meanwhile, the application of TREESYNTH-guided data balance leads to performance improvements of 7.9% and 2.9% for Temperature Sampling and Persona Hub, respectively. To demonstrate the underlying data distribution more intuitively, Figure 7 presents the t-SNE visualization of data spatial distribution for different approaches. Benefiting from the global perspective, TREESYNTH exhibits comprehensive and well-balanced distribution across the data space. In contrast, samples from Temperature Sampling and Persona Hub predominantly tend to cluster in limited subspaces, revealing insufficient diversity. The application of TREESYNTH substantially enhances their distributional uniformity, enabling comprehensive coverage of the data space. These results collectively demonstrate that TREESYNTH-guided data balance effectively addresses deficiencies in existing datasets by optimizing their sample distribution, leading to measurable improvements in downstream model performance.

---

[5]Evol-Instruct is excluded in this section, due to its strong dependence on seed dataset.

# 5 Related Work

**Data Synthesis via LLMs.** Owing to their remarkable ability to generate large-scale, high-quality datasets, LLMs are increasingly being explored as an effective alternative to the time-consuming and labor-intensive process of human annotation [28–30]. Pioneering studies [31, 18, 32] have showcased the use of LLMs to paraphrase or augment existing instruction datasets into more comprehensive ones. Furthermore, recent research has explored data synthesis from scratch in areas such as mathematics [15], code [33], general alignment [34], etc. However, uncontrolled synthesis process often exhibits significant biases, favoring easy queries while neglecting more challenging ones [35–37]. To enhance controllability, Wong et al. [38] and Huang et al. [39] incorporate strata and topic clustering as guidance, respectively. Our approach distinguishes itself by constructing a tree-like hierarchical structure from scratch to model the data space for better controllability.

**Diversity Enhancement in Data Synthesis.** The diversity of training datasets is essential for enhancing model generalization [40]. Numerous studies have sought to improve diversity during data synthesis [41–43]. Specifically, increasing sampling temperature [16] can generate more diverse data; however, it often provides limited domain coverage and decreased quality [44]. To promote diversity while ensuring quality, recent methods take advantage of the in-context learning capabilities of LLMs through manually designed rules for data evolution [18, 45–48]. However, from the perspective of data space, they typically start from the local distribution (*i.e.*, seed data) without the global view, hindering their comprehensive coverage. Besides, synthesized data often becomes repetitive and homogeneous with the increase of data scales due to the inherent model biases. To address this challenge, various competing methods have emerged, particularly those based on attribute combinations [49–52], such as persona-driven data synthesis [17]. However, these approaches typically rely on fixed attribute dimensions curated by LLMs or human knowledge. In contrast, our work emphasizes the dynamic construction of a tree structure to iteratively partition the entire domain space, ensuring comprehensive coverage without human intervention.

**Application of Tree Structure.** As a canonical data structure, trees have been applied in various machine learning algorithms. Specifically, decision trees conceptualize discriminative tasks as a search problem through a tree-like combinatorial problem space [53–55]. Besides, Hao et al. [56], Yao et al. [57] utilize tree-search algorithms, including breadth-first search, depth-first search, and Monte Carlo Tree Search [58], to guide multi-step reasoning process for improved reasoning capabilities of LLMs. Furthermore, AlphaZero-like tree search learning approach [59] leverages a learned value function to guide the decoding process of LLMs, and particularly improves the performance on long-horizon planning. In contrast, our method does not rely on tree structures for discrimination tasks or to improve reasoning capabilities. Instead, it focuses on data synthesis through hierarchical tree-like spatial partitioning. Besides, the decision rules are not learned but are directly derived from the extensive knowledge inherent in LLMs.

# 6 Conclusion

Targeting synthesizing diverse datasets with LLMs from scratch, we propose a tree-guided subspace-based data synthesis approach, TREESYNTH, which recursively partitions a task-specific full data space into mutually exclusive and exhaustive atomic subspaces before synthesizing and collecting subspace samples into a comprehensive dataset. This globally divide-and-synthesize strategy effectively circumvents repetition and space collapse caused by model biases, seed data, and low-variation prompts in prior methods, promoting the diversity of large-scale data synthesis. Besides, TREESYNTH also facilitates sample allocation into atomic subspaces, enabling re-balancing of existing datasets for more balanced distributions. Compared with both human-crafted datasets and peer data synthesis methods, TREESYNTH consistently achieves superior data diversity, model performance, robust scalability, and data balance efficacy, establishing itself as a promising solution for diverse data synthesis without seed data.

# 7 Acknowledgement

This work was supported in part by Hong Kong Innovation and Technology Commission's Innovation and Technology Fund (Award No. ITS/269/22FP), Hong Kong RGC grants C7004-22G (CRF) and CRS_PolyU501/23 (CRS).

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

# A Appendix

## A.1 Pseudo Code of TREESYNTH

The pseudo code of TREESYNTH is presented in Algorithm 1.

## A.2 Benchmarks

To thoroughly evaluate the performance improvements enabled by TREESYNTH-generated data, we test models across three domains: mathematical reasoning, code generation, and psychology tasks. The mathematical reasoning tasks use the MATH dataset test set [21] and GSM8K benchmark [20]. Code generation capabilities are assessed through HumanEval [23] and MBPP [22]. For psychology tasks, we employ the SimpleToM dataset [24]. These benchmarks collectively assess our method's effectiveness across multiple dimensions, with brief descriptions provided below:

- **GSM8K** evaluates mathematical reasoning capabilities through 8,500 high-quality grade school math problems developed via human expert annotation. The dataset is partitioned into 7,500 training and 1,000 test problems, each requiring 2-8 computational steps combining basic arithmetic operations.

- **MATH** dataset comprises 12,500 competition-level mathematics problems, with 7,500 designated for training and 5,000 for testing. It requires step-by-step solutions, emphasizing accurate problem decomposition and the generation of formal proofs for multi-step mathematical challenges.

- **HumanEval** evaluates functional correctness through hand-written programming problems requiring language comprehension, reasoning, algorithms, and mathematics. It comprises 164 problems with function signatures, docstrings, and unit tests (avg. 7.7 per problem). Tasks are manually crafted to avoid data leakage from public code repositories.

- **MBPP** evaluates programming competence through entry-level Python functions requiring implementation based on textual specifications and test case verification. It contains 974 crowd-sourced programming problems with functional correctness validation, including a core subset of author-curated solutions with manual quality assurance.

- **SimpleToM** is designed to evaluate whether LLMs can implicitly apply Theory of Mind (ToM) reasoning to predict behavior and judge the rationality of observed actions in social scenarios. It consists of concise, diverse stories followed by three questions that assess different levels of ToM reasoning: predicting a character's mental state, forecasting their behavior, and judging the rationality of their actions[6].

## A.3 Baselines

The concise descriptions of all the baselines are presented below.

- **Vanilla Data.** "Vanilla Data" refers to the original GSM8K and MATH training sets, as well as the Code Alpaca Python subset used for HumanEval and MBPP, with details on GSM8K and MATH datasets provided in Sec A.2. The Code Alpaca dataset is a collection of 20,000 instruction-following data points, each containing a unique task description, optional input context, and a corresponding output, designed to train a language model for code generation tasks by following specific instructions. Only Python-related samples are retained, aligning with HumanEval and MBPP.

- **Temperature Sampling.** Temperature sampling adjusts the softmax function of LLMs to control output randomness by scaling logits. Higher temperatures increase creativity and randomness, while lower temperatures result in more deterministic outputs. Following the prompts consistent with TREESYNTH, as illustrated in Figures 13, 14, and 15, we set the temperature to 0.7 and generate 100k baseline data in GSM8K, MATH, and Code

---

[6]In the SimpleTom-style dataset, data where the protagonist is unaware of hidden information is labeled as negative samples, while data where the protagonist is aware of hidden information is labeled as positive samples. In this paper, all SimpleToM-style data starts by generating negative samples to account for half of the dataset. These are then converted into positive samples using the prompts shown in Figures 29. The combination of positive and negative samples forms a complete dataset.

---

**Algorithm 1** Pseudo Code of TREESYNTH

---

**Require:** Context description $\mathcal{C}_\mathcal{O}$ of entire data space $\mathcal{O}$, Pivot sample count $l$, Maximum attribute value count $N$, Maximum tree depth $d$

1: **function** CRITERION_DETERMINATION($\mathcal{C}_\mathcal{S}, l$)
2:     Generate diverse pivot samples $\mathcal{X} = \{x_t\}_{t=1}^l$ via $\text{LLM}_{\text{pivot}}(\mathcal{C}_\mathcal{S})$
3:     Determine exactly one core criterion $\delta \leftarrow \text{LLM}_{\text{determine}}(\mathcal{X})$
4:     Obtain mutually exclusive attribute values $V_\mathcal{X}^\delta \leftarrow \texttt{Partition}(\mathcal{X}, \delta)$
5:     **return** $\delta, V_\mathcal{X}^\delta$
6: **end function**
7: **function** SUBSPACE_COVERAGE($\mathcal{C}_\mathcal{S}, \delta, V_\mathcal{X}^\delta, N$)
8:     Fully cover the criterion $\delta$ with all the attribute values $V_\mathcal{S} \leftarrow \text{LLM}_{\text{complement}}(V_\mathcal{X}, \delta)$
9:     **if** $|V_\mathcal{S}| > N$ **then**
10:         Create subspace $\mathcal{S}_{\text{sub}}^{\inf}(V_\mathcal{S})$                    ▷ Randomly sample an attribute value whenever needed.
11:     **else**
12:         Create subspace $\mathcal{S}_{\text{sub}}^i$ for each $v_i \in V_\mathcal{S}$
13:     **end if**
14:     **return** $\mathcal{S}_{\text{sub}}^*$
15: **end function**

      ▷ Stage 1: Data Space Partitioning
16: Initialize root node $\mathcal{N}_{Root}$ (*i.e.*, $\mathcal{O}$ with $\mathcal{C}_\mathcal{O}$) with the depth as 0, and BFS queue $Q \leftarrow \{\mathcal{N}_{Root}\}$
17: **while** $Q \neq \varnothing$ **do**
18:     Dequeue the first node $\mathcal{N}'$ from $Q$, and obtain its depth $d'$ and context description $\mathcal{C}_{\mathcal{N}'}$
19:     **if** $d' < d$ **then**
20:         $\delta, V_\mathcal{X}^\delta \leftarrow$ CRITERION_DETERMINATION($\mathcal{C}_{\mathcal{N}'}, l$)                    ▷ Step 1: Criterion Determination
21:         $\mathcal{N}_{\text{sub}}'^* \leftarrow$ SUBSPACE_COVERAGE($\mathcal{C}_{\mathcal{N}'}, \delta, V_\mathcal{X}^\delta, N$)                    ▷ Step 2: Subspace Coverage
22:         Add $\mathcal{N}_{\text{sub}}'^*$ as the child nodes of $\mathcal{N}'$, and append to the queue $Q \leftarrow Q \cup \mathcal{N}_{\text{sub}}'^*$
23:     **else**
24:         Mark $\mathcal{N}'$ as a leaf node $\mathcal{N}_{\text{Leaf}}'$
25:     **end if**
26: **end while**
        **return** Spatial partitioning tree $\mathcal{T}$ with the root node $\mathcal{O}$

      ▷ Stage 2: Subspace Data Synthesis
27: Collect all leaf nodes $\mathcal{N}_{\text{Leaf}}^* \leftarrow \{\mathcal{N}_{\text{Leaf}}^1, \mathcal{N}_{\text{Leaf}}^2, \cdots\}$
28: **for** each $\mathcal{N}_{\text{Leaf}}^i$ in $\mathcal{N}_{\text{Leaf}}^*$ **do**
29:     Trace hierarchical parent nodes from the root node $\mathcal{P} \leftarrow \{\mathcal{N}_{Root}, \cdots, \mathcal{N}_{\text{Leaf}}^i\}$
30:     Obtain the attribute intersections as the leaf node description $\mathcal{C}_{\mathcal{N}_{\text{Leaf}}^i} \leftarrow \bigcap_{j \in \mathcal{P}} V_j$
31:     Generate diverse leaf samples $\mathcal{D}_{\text{Leaf}}^i = \{x_k\}_{k=1}^N$ via $\text{LLM}_{\text{sample}}(\mathcal{C}_{\mathcal{N}_{\text{Leaf}}^i})$
32: **end for**
33: Collect all the leaf samples into the final dataset $\mathcal{D}_{\text{final}} \leftarrow \bigcup \mathcal{D}_{\text{Leaf}}^*$
        **return** $\mathcal{D}_{\text{final}}$                    ▷ with high diversity, good balance, and comprehensive coverage.

      ▷ TREESYNTH-Guided Data Balance
**Require:** Dataset $\mathcal{D} = \{d_u | u = 1, 2, ..., w\}$, data description $\mathcal{C}_\mathcal{D}$, threshold $N_{\text{Sub}}$
34: Build tree $\mathcal{T}_\mathcal{D}$ via Data Space Partitioning using $\mathcal{C}_\mathcal{D}$
35: Initialize leaves $\mathcal{L} \leftarrow \{\mathcal{N}_{\text{Leaf}}^1, ..., \mathcal{N}_{\text{Leaf}}^m\}$
36: **for** $d_u \in \mathcal{D}$ **do**
37:     Route $d_u$ to leaf $\mathcal{N}_{\text{Leaf}}^k$ via $\mathcal{T}_\mathcal{D}$
38: **end for**
39: **for** $\mathcal{N}_{\text{Leaf}}^i \in \mathcal{L}$ **do**
40:     $\mathcal{D}^i \leftarrow$ samples in $\mathcal{N}_{\text{Leaf}}^i$
41:     **if** $|\mathcal{D}^i| > N_{\text{Sub}}$ **then**
42:         $D_{\text{new}}^i \leftarrow$ Downsample to $N_{\text{Sub}}$
43:     **else if** $|\mathcal{D}^i| < N_{\text{Sub}}$ **then**
44:         $D_{\text{new}}^i \leftarrow$ Synthesize $(N_{\text{Sub}} - |\mathcal{D}^i|)$ samples in $\mathcal{N}_{\text{Leaf}}^i$ via LLMs and add into $D^i$
45:     **end if**
46: **end for**
47: Aggregate $\mathcal{D}_{\text{balance}} \leftarrow \bigcup \mathcal{D}_{\text{new}}^i$ **return** $\mathcal{D}_{\text{balance}}$

---

Alpaca styles, respectively. During the construction of the SimpleToM-style training set, the temperature parameter is set to 0.7 while generating 20k data samples through the combined use of Figures 16 and 29.

- **Evol-Instruct.** Evol-Instruct is a method that uses LLMs to automatically generate diverse and complex instruction datasets by evolving initial instructions through in-depth and in-breadth processes, enhancing the complexity and diversity of instructions for training LLMs. This paper utilizes Evol-Instruct to enhance the training datasets of GSM8K, MATH, and Code Alpaca, constructing 100k samples for each respective style (GSM8K, MATH, Code Alpaca) through multiple evolutionary iterations.

- **Persona Hub.** Persona Hub consists of 1 billion diverse personas curated from web data, enhancing diversity in large-scale data synthesis when incorporated into synthetic prompts. By randomly sampling personas from Persona Hub to replace the original "math expert" and "coding expert" profiles in Figures 13, 14, and 15, and by prepending personas to the prompts in Figure 16, we synthesize 100k samples each for GSM8K, MATH, and Code Alpaca-style datasets, as well as 20k SimpleToM-style data through this persona substitution approach.

## A.4 Experiments Details

The training dataset is constructed through TREESYNTH-generated instructions, with LLMs subsequently producing answers corresponding to these instructions.

**Spatial Partitioning Tree Construction.** For each task, as illustrated in Figure 17, 18, 19 and 20, we develop training data descriptions $\mathcal{C}_{GSM8K}$, $\mathcal{C}_{MATH}$, $\mathcal{C}_{Code}$ and $\mathcal{C}_{ToM}$ following the standards of GSM8K, MATH, Code Alpaca and SimpleToM respectively. The prompt words for Criterion Determination and Subspace Coverage are presented in Figure 21, 22, 23, 24 and 25, 26, 27, 28 respectively. Afterwards, we construct the spatial partitioning trees $\mathcal{T}_{GSM8K}$, $\mathcal{T}_{MATH}$ and $\mathcal{T}_{Code}$ using GPT-4o, LLaMA3.3-70B-Instruct and Qwen2.5-72B-Instruct with maximum tree depth $d$ set to 4, pivot count $l$ to 10, and maximum attribute values $N$ to 50. For SimpleToM-style data, we employ the LLaMA3.3-70B-Instruct model with a maximum tree depth $d = 3$, while maintaining consistency in all other parameters with the configurations of other tasks, to generate the spatial partitioning tree $\mathcal{T}_{ToM}$.

**Training Data Generation.** For each leaf node in $\mathcal{T}_{GSM8K}$, $\mathcal{T}_{MATH}$ and $\mathcal{T}_{Code}$, we generate $N_{\text{Leaf}} = 10$ data instances using the prompts from Figures 17, 18, and 19 within their corresponding subspaces to construct the raw instruction sets using GPT-4o, LLaMA3.3-70B-Instruct and Qwen2.5-72B-Instruct. For coding-task instructions, following the practice of Code Alpaca [26], we compute pairwise Rouge [60] similarity scores between all data entries and filter out data pairs with similarity scores exceeding 0.7. Subsequently, we randomly select 100k instructions from the raw instruction sets and use the same LLMs that generated the instructions to produce answers, forming the training dataset. For the generation of SimpleToM-style data, we use the prompt from Figure 20 command LLaMA3.3-70B-Instruct to generate $N_{\text{Leaf}} = 10$ instructions for each leaf node of $\mathcal{T}_{ToM}$, forming an instruction set containing 20k samples. We then use LLaMA3.3-70B-Instruct to generate corresponding answers for these instructions.

**TREESYNTH-Guided Data Balance.** We perform a TREESYNTH-guided data balancing strategy for SimpleToM-style datasets. Using the prompt templates from Figure 30, we systematically partition both Temperature Sampling and Persona Hub-style SimpleToM data into distinct subspaces of $\mathcal{T}_{ToM}$. The sample threshold $N_{Sub}$ for each subspace is set to 10 to achieve data balancing across these subspaces.

**Model Training.** To fine-tune our selected base models (*i.e.*, LLaMA3.1-8B and Qwen2.5-7B), we employ the parameter-efficient fine-tuning method **LoRA** [61–63]. Specifically, we uniformly set the lora_dropout = 0, weight_decay = 0.1, and trained each model for 5 epochs. For GSM8K-style data, we set the learning rate to $1 \times 10^{-4}$. For MATH, Code Alpaca, and SimpleToM-style data, we set

the learning rate to $1 \times 10^{-5}$ during training [7]. Empirical observations show that these configurations consistently achieves stable and competitive downstream performance across various tasks.

### A.5 Complementary Analysis

**t-SNE visualization of SimpleToM-style datasets.** As shown in Figure 7, we illustrate t-SNE distributions of SimpleToM-style datasets synthesized by various methods alongside their TREESYNTH-guided data-balanced counterparts, demonstrating significantly improved and more comprehensive coverage.

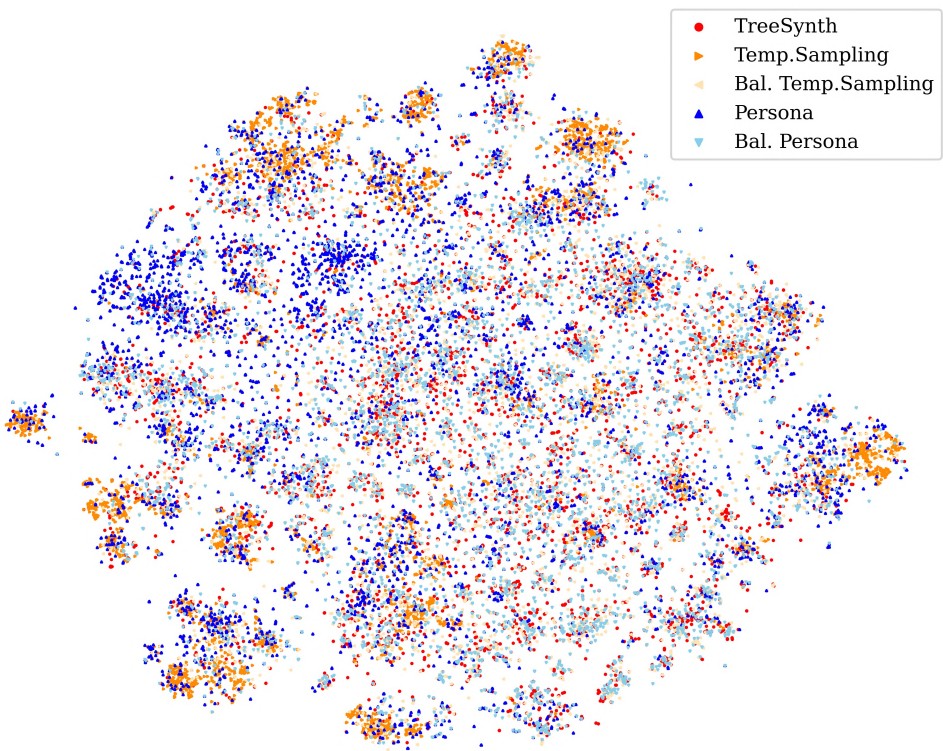

Figure 7: t-SNE visualization of SimpleToM-style datasets synthesized by different methods and their TREESYNTH-guided data balanced counterparts, exhibiting remarkably comprehensive coverage.

**Model performance and data diversity comparison.** The results presented in Tables 4 demonstrates a comparison of data diversity and model performance for different methods powered by `Qwen2.5-72B-Instruct` across multiple benchmarks, highlighting TREESYNTH's superior data diversity and robust performance improvement in downstream tasks.

**Model performance across data scales.** The detailed numerical values of scalability evaluations across GSM8K, MATH, MBPP and HumanEval benchmarks are provided in Table 5, 6, 7 and 8, respectively. In addition, Figure 8 displays the performance trends of various methods using the `Qwen2.5-72B-Instruct` generation model across different data scales. TREESYNTH consistently outperforms all baselines while exhibiting near-linear scaling with data growth, demonstrating its superior scalability.

**Broadening TREESYNTH Assessment Beyond LLaMA and Qwen.** To further validate the robustness and generalizability of TREESYNTH, we extend our assessment to `Gemma-3-PT-4B` and `Gemma-3-PT-12B`, covering additional model families and sizes with all data synthesis methods

---

[7]These hyperparameters are selected based on a comprehensive grid search over candidate values: `learning rate` $\in \{1 \times 10^{-6}, 5 \times 10^{-6}, 1 \times 10^{-5}, 5 \times 10^{-5}, 1 \times 10^{-4}, 5 \times 10^{-4}\}$, `lora_dropout` $\in \{0, 0.05\}$, `weight_decay` $\in \{0, 0.1\}$, and `epoch count` $\in \{3, 5, 7, 10\}$.

| Method | GSM8K↑ | Diversity↓ | MATH↑ | Diversity↓ | MBPP↑ | HumanEval↑ | Diversity↓ | Avg.↑ |
|---|---|---|---|---|---|---|---|---|
| *Foundation Model:* LLAMA-3.1 8B | | | | | | | | |
| Zero-Shot | 4.85 | - | 3.54 | - | 19.8 | 15.85 | - | 11.01 |
| Few-Shot | 40.26 | - | 20.46 | - | - | - | - | - |
| Vanilla Data | 58.15 | 0.40 | 19.48 | 0.16 | 46.8 | 43.29 | 0.29 | 41.93 |
| Temp. Sampling | 64.75 | 0.41 | 28.14 | 0.21 | 43.2 | 42.68 | 0.31 | 44.69 |
| Evol-Instruct | 66.72 | **0.36** | 30.52 | 0.19 | 39.8 | 42.07 | 0.27 | 44.78 |
| Persona Hub | 61.71 | 0.38 | 28.12 | 0.20 | 42.8 | 42.07 | 0.28 | 43.67 |
| TREESYNTH | **68.31** | 0.36 | **31.14** | **0.15** | **47.4** | **48.17** | **0.22** | **48.76** |
| *Foundation Model:* QWEN-2.5 7B | | | | | | | | |
| Zero-Shot | 54.97 | - | 54.38 | - | 11.2 | 54.88 | - | 43.86 |
| Few-Shot | 67.40 | - | 47.58 | - | - | - | - | - |
| Vanilla Data | 68.76 | 0.40 | 47.68 | 0.16 | 53.4 | 77.44 | 0.29 | 61.82 |
| Temp. Sampling | 77.26 | 0.41 | 62.08 | 0.21 | 53.2 | 78.05 | 0.31 | 67.65 |
| Evol-Instruct | 78.70 | **0.36** | 67.56 | 0.19 | 57.8 | 76.22 | 0.27 | 70.07 |
| Persona Hub | 79.15 | 0.38 | 61.98 | 0.20 | 57.2 | 76.22 | 0.28 | 68.64 |
| TREESYNTH | **84.99** | 0.36 | **68.44** | **0.15** | **61.4** | **78.66** | **0.22** | **73.37** |

Table 4: Model performance and data diversity comparison of various methods with `Qwen2.5-72B-Instruct`-powered data synthesis across two foundation models and multiple benchmarks. "Zero-Shot" and "Few-Shot" exhibit the base performance of foundation models. "Temp. Sampling" is abbreviated from "Temperature Sampling". "Vanilla Data" denotes the original GSM8K and MATH training sets, and the Code Alpaca Python subset for HumanEval and MBPP. "Diversity" is measured by cosine similarity, where lower values indicate greater diversity. Bold and underlining indicate the best and second-best indicators, respectively. "Avg." means the average of the performance scores across all the benchmarks.

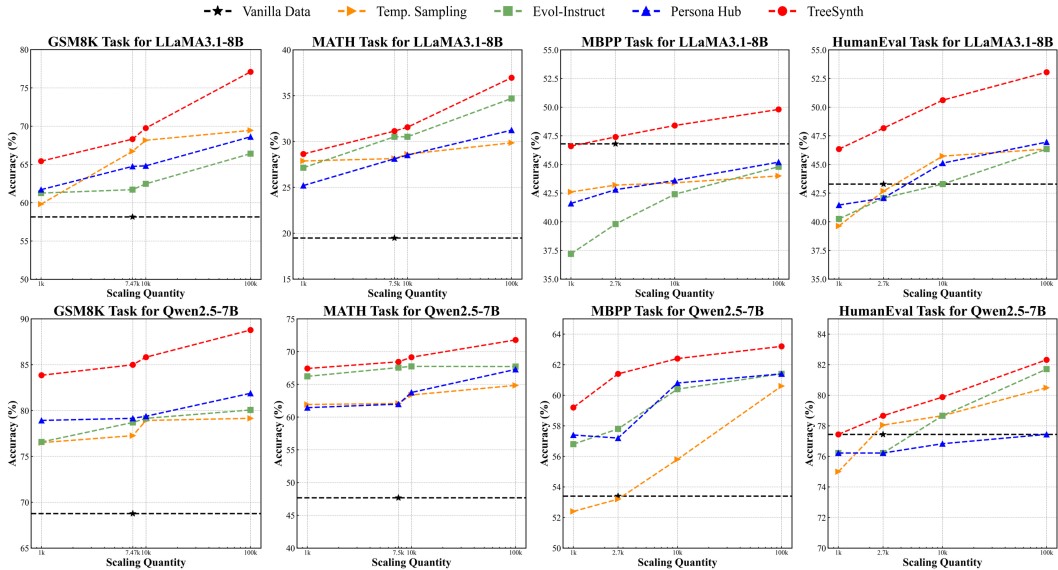

Figure 8: Model performance trends across data scales for different methods powered by `Qwen2.5-72B-Instruct`. "Temp. Sampling" refers to Temperature Sampling. "Vanilla Data" denotes original GSM8K and MATH training sets, and Code Alpaca Python subset for HumanEval and MBPP.

powered by `GPT-4o`, as shown in Tables 9, 10 and Figure 9. The trends mirror those reported in Sec. 4.2: TREESYNTH consistently surpasses human-annotated data and peer synthesis methods, while its downstream performance consistently scales with data volumes. This strongly validates its robustness and generalizability.

**Computational Costs of TREESYNTH.** TREESYNTH can be powered by open-source models, which ensures that its operational cost and overhead are extremely low. As detailed in Table 11,

| Model | Gen. Model | Dataset Scaling | @1k | @7k | @10k | @100k |
|---|---|---|---|---|---|---|
| LLaMA3.1-8B | GSM8K training set | Vanilla Data | - | 58.15 | - | - |
| | GPT-4o | Evol-Instruct | 56.86 | 61.03 | 63.38 | 66.64 |
| | | Persona Hub | 62.47 | 63.38 | 64.22 | 67.78 |
| | | Temp. Sampling | 54.89 | 54.97 | 56.94 | 56.79 |
| | | **TREESYNTH** | **64.29** | **66.72** | **69.75** | **74.07** |
| | LLaMA3.3-70B-Instruct | Evol-Instruct | 59.74 | 63.46 | 66.41 | 68.31 |
| | | Persona Hub | 62.70 | 61.41 | 63.76 | 68.92 |
| | | Temp. Sampling | 53.15 | 55.42 | 57.39 | 56.79 |
| | | **TREESYNTH** | **68.84** | **69.45** | **71.95** | **77.79** |
| | Qwen2.5-72B-Instruct | Evol-Instruct | 59.82 | 66.72 | 68.16 | 69.45 |
| | | Persona Hub | 61.26 | 61.71 | 62.47 | 66.41 |
| | | Temp. Sampling | 61.71 | 64.75 | 64.82 | 68.61 |
| | | **TREESYNTH** | **65.43** | **68.31** | **69.75** | **77.10** |
| Qwen2.5-7B | GSM8K training set | Vanilla Data | - | 68.76 | - | - |
| | GPT-4o | Evol-Instruct | 73.77 | 73.16 | 77.94 | 79.38 |
| | | Persona Hub | 81.88 | 83.24 | 83.78 | 83.09 |
| | | Temp. Sampling | 80.74 | 80.67 | 81.35 | 82.49 |
| | | **TREESYNTH** | **84.76** | **86.13** | **86.35** | **87.49** |
| | LLaMA3.3-70B-Instruct | Evol-Instruct | 74.15 | 75.13 | 75.66 | 77.26 |
| | | Persona Hub | 81.73 | 82.79 | 83.70 | 84.15 |
| | | Temp. Sampling | 61.49 | 69.67 | 69.75 | 78.39 |
| | | **TREESYNTH** | **84.46** | **85.44** | **85.82** | **86.81** |
| | Qwen2.5-72B-Instruct | Evol-Instruct | 76.57 | 78.70 | 79.15 | 80.06 |
| | | Persona Hub | 78.92 | 79.15 | 79.38 | 81.88 |
| | | Temp. Sampling | 76.50 | 77.26 | 78.92 | 79.15 |
| | | **TREESYNTH** | **83.85** | **84.99** | **85.82** | **88.78** |

Table 5: Comparison of instruction-tuned model performance on GSM8K using training data from different sources and generation methods. For each data scale (1k, 7k, 10k, 100k), models are fine-tuned on equally sized datasets constructed via various data synthesis methods, as well as the original GSM8K training set. "Temp. Sampling" is abbreviated from "Temperature Sampling". "Vanilla Data" denotes the original GSM8K training sets.

we calculate token consumption when synthesizing 100k-scale datasets of various styles using TREESYNTH driven by LLaMA3.1-70B-Instruct. Based on the API pricing[8] (0.038/0.12 per million input/output tokens), we estimate that the cost to synthesize each dataset is less than 10 USD. Moreover, the efficiency of data synthesis should be considered in tandem with downstream training costs. Specifically, synthetic data requires model training to be converted into performance gains. High-quality data can lead to more significant savings in model training, whose cost is markedly higher than the inference expense for data synthesis. Therefore, enhancing data quality is a more strategic approach to significantly reduce the greater expense of downstream training. As shown in Figures 5, 6, and 8, TREESYNTH requires only ~1k samples to match or exceed the downstream performance of 100k-sample baselines, precisely highlighting its value and efficiency.

**Extensive Evaluation of Generation Quality.** As described in Sec. A.4, we empirically set the tree depth to 4. Table 12 provides statistics on the number of criteria and attribute values at each depth within the spatial partitioning trees that TREESYNTH generates for various tasks. We also present the subtree of MATH in Figure 10, with the further partitioning criteria denoted in brackets. The results indicate that as depth increases, the quantities of both criteria and attribute values grow exponentially. This demonstrates that by deepening its structure, TREESYNTH continuously partitions the data space and describes the resulting subspaces with more fine-grained attribute values. This capability forms the basis of TREESYNTH's superior performance in data diversity, downstream task performance, and scalability.

---

[8]https://deepinfra.com/meta-llama/Llama-3.3-70B-Instruct-Turbo

| Model | Gen. Model | Dataset Scaling | @1k | @7.5k | @10k | @100k |
|-------|------------|-----------------|-----|-------|------|-------|
| LLaMA3.1-8B | MATH training set | Vanilla Data | - | 19.48 | - | - |
| | GPT-4o | Evol-Instruct | 23.74 | 24.58 | 25.16 | 29.86 |
| | | Persona Hub | 25.60 | 27.74 | 28.20 | 29.62 |
| | | Temp. Sampling | 24.34 | 24.28 | 25.04 | 26.86 |
| | | **TREESYNTH** | **28.02** | **30.34** | **31.34** | **40.10** |
| | LLaMA3.3-70B-Instruct | Evol-Instruct | **26.78** | 27.26 | 28.26 | 31.24 |
| | | Persona Hub | 23.52 | 23.78 | 23.52 | 24.60 |
| | | Temp. Sampling | 21.84 | 22.08 | 23.18 | 22.70 |
| | | **TREESYNTH** | 26.68 | **27.52** | **28.94** | **32.42** |
| | Qwen2.5-72B-Instruct | Evol-Instruct | 27.14 | 30.52 | 30.52 | 34.70 |
| | | Persona Hub | 25.20 | 28.12 | 28.52 | 31.24 |
| | | Temp. Sampling | 27.88 | 28.14 | 28.62 | 29.86 |
| | | **TREESYNTH** | **28.64** | **31.14** | **31.56** | **36.96** |
| Qwen2.5-7B | MATH training set | Vanilla Data | - | 47.68 | - | - |
| | GPT-4o | Evol-Instruct | 58.52 | 61.10 | 61.20 | 64.84 |
| | | Persona Hub | 63.62 | 66.22 | 67.42 | 67.72 |
| | | Temp. Sampling | 63.38 | 62.76 | 63.36 | 61.20 |
| | | **TREESYNTH** | **64.48** | **66.84** | **67.74** | **69.90** |
| | LLaMA3.3-70B-Instruct | Evol-Instruct | 57.30 | 59.60 | 58.56 | 61.98 |
| | | Persona Hub | 60.92 | 61.98 | 62.06 | 63.12 |
| | | Temp. Sampling | 61.64 | 61.70 | 61.96 | 62.88 |
| | | **TREESYNTH** | **62.08** | **63.28** | **64.84** | **66.80** |
| | Qwen2.5-72B-Instruct | Evol-Instruct | 66.22 | 67.56 | 67.74 | 67.72 |
| | | Persona Hub | 61.44 | 61.98 | 63.76 | 67.28 |
| | | Temp. Sampling | 61.92 | 62.08 | 63.38 | 64.84 |
| | | **TREESYNTH** | **67.42** | **68.44** | **69.14** | **71.78** |

Table 6: Comparison of instruction-tuned model performance on MATH using training data from different sources and generation methods. For each data scale (1k, 7.5k, 10k, 100k), models are fine-tuned on equally sized datasets constructed via various data synthesis methods, as well as the original MATH training set. "Temp. Sampling" is abbreviated from "Temperature Sampling". "Vanilla Data" denotes the original MATH training sets.

**Qualitative Analysis for TREESYNTH.** Training Qwen-2.5-7B on MATH-style data from either TREESYNTH or Persona Hub reveals that TREESYNTH yields consistent gains across all seven sub-domains, with pronounced improvements in Number Theory, Geometry, and Counting & Probability, as shown in Table 13. A specific case in Number Theory, detailed in Figure 11, further highlights this superiority where the TREESYNTH-trained LLM correctly computes modular inverses while the Persona Hub-trained model fails. This contrast suggests that specialized fields demanding precise computations, such as modular arithmetic, may expose data biases in Persona Hub. Overall, the comprehensive and substantial improvements provide empirical evidence that TREESYNTH's data generation strategy achieves superior diversity and more effective data space coverage, enabling the model to attain markedly better performance in specialized mathematical reasoning.

**Validation of the Criterion Selection Mechanism.** Prompting the LLM to approximate this objective is feasible with verified effectiveness. As shown in Figures 21–24, we explicitly instruct the LLM to "Identify exactly ONE core dimension that best distinguishes pivot samples". Given the powerful capabilities of modern LLMs, they generally provide a satisfactory criterion. As shown in Figure 12, given five MATH-style geometry problems, the LLM can identify `geometric_object_type` as the criterion to effectively partition the data.

## A.6 Case Study

**The data generation process of TREESYNTH demonstrates both high controllability and interpretability.** As illustrated in Figures 31, 32, and 33, the GSM8K, MATH, and Code Alpaca-style

| Model | Gen. Model | Dataset Scaling | @1k | @2k | @10k | @100k |
|---|---|---|---|---|---|---|
| LLaMA3.1-8B | Code Alpaca Python subset | Vanilla Data | - | 46.8 | - | - |
| | GPT-4o | Evol-Instruct | 40.8 | 45.2 | 45.8 | 47.2 |
| | | Persona Hub | 42.0 | 45.2 | 45.6 | 47.8 |
| | | Temp. Sampling | 43.4 | 44.8 | 47.8 | 46.8 |
| | | TREESYNTH | **49.4** | **50.8** | **52.6** | **54.4** |
| | LLaMA3.3-70B-Instruct | Evol-Instruct | 39.0 | 40.6 | 43.4 | 46.6 |
| | | Persona Hub | 46.6 | 45.8 | 46.8 | 48.4 |
| | | Temp. Sampling | 44.4 | 44.6 | 45.8 | 46.2 |
| | | TREESYNTH | **47.8** | **50.2** | **50.8** | **52.0** |
| | Qwen2.5-72B-Instruct | Evol-Instruct | 37.2 | 39.8 | 42.4 | 44.8 |
| | | Persona Hub | 41.6 | 42.8 | 43.6 | 45.2 |
| | | Temp. Sampling | 42.6 | 43.2 | 43.4 | 44.0 |
| | | TREESYNTH | **45.8** | **46.6** | **47.4** | **49.8** |
| Qwen2.5-7B | Code Alpaca Python subset | Vanilla Data | - | 53.4 | - | - |
| | GPT-4o | Evol-Instruct | 57.2 | 59.2 | 60.6 | 61.6 |
| | | Persona Hub | 58.8 | 61.6 | 61.0 | 62.2 |
| | | Temp. Sampling | 60.0 | 59.6 | 59.0 | 60.6 |
| | | TREESYNTH | **61.2** | **62.8** | **63.2** | **64.8** |
| | LLaMA3.3-70B-Instruct | Evol-Instruct | 54.6 | 55.8 | 57.4 | 58.6 |
| | | Persona Hub | 56.0 | 58.8 | 59.4 | 61.8 |
| | | Temp. Sampling | 53.8 | 54.8 | 56.0 | 59.6 |
| | | TREESYNTH | **57.6** | **59.4** | **61.4** | **63.0** |
| | Qwen2.5-72B-Instruct | Evol-Instruct | 56.8 | 57.8 | 60.4 | 61.4 |
| | | Persona Hub | 57.4 | 57.2 | 60.8 | 61.4 |
| | | Temp. Sampling | 52.4 | 53.2 | 55.8 | 60.6 |
| | | TREESYNTH | **59.2** | **61.4** | **62.4** | **63.2** |

Table 7: Comparison of instruction-tuned model performance on MBPP using training data from different sources and generation methods. For each data scale (1k, 2k, 10k, 100k), models are fine-tuned on equally sized datasets constructed via various data synthesis methods, as well as the common training set for coding task. "Temp. Sampling" is abbreviated from "Temperature Sampling". "Vanilla Data" denotes the Code Alpaca training sets.

datasets generated by TREESYNTH are presented alongside their corresponding criteria and attribute values. The generated data not only preserves the stylistic features of the original datasets but also strictly adheres to specified attribute values. This highlights TREESYNTH's key strength: by leveraging attribute values associated with subspaces obtained through data space partitioning, the approach achieves precise control over data generation within each subspace, thereby ensuring effective regulation and interpretability of the synthesis process.

## A.7 Limitations

This paper introduces TREESYNTH, a tree-guided subspace-based synthesis approach that systematically partitions the data space from a global perspective to produce diverse and comprehensive instruction sets. After generating these instructions, LLMs are utilized to synthesize corresponding answers. However, following the practice of Evol-Instruct [18], the accuracy validation of answers remains unexplored. Despite this potential defect, TREESYNTH still demonstrates consistent improvements on both data diversity and downstream task performance, highlighting its superior effectiveness and robust scalability.

## A.8 Experiments Compute Resources

All experiments are executed on high-performance computing node equipped with eight NVIDIA H100 SXM GPUs (80 GB HBM3 each), dual-socket 128-core CPUs, and 2 TB of system RAM.

| Model | Gen. Model | Dataset Scaling | @1k | @2k | @10k | @100k |
|---|---|---|---|---|---|---|
| LLaMA3.1-8B | Code Alpaca Python subset | Vanilla Data | - | 43.29 | - | - |
| | GPT-4o | Evol-Instruct | 44.51 | 49.39 | 51.22 | 51.22 |
| | | Persona Hub | 43.90 | 45.12 | 48.78 | 49.39 |
| | | Temp. Sampling | 41.46 | 45.73 | 46.95 | 47.56 |
| | | TREESYNTH | **48.78** | **50.00** | **51.83** | **53.05** |
| | LLaMA3.3-70B-Instruct | Evol-Instruct | 40.85 | 41.46 | 42.68 | 45.12 |
| | | Persona Hub | 39.63 | 40.24 | 44.51 | 46.34 |
| | | Temp. Sampling | 40.24 | 41.46 | 42.68 | 45.12 |
| | | TREESYNTH | **47.56** | **48.17** | **50.00** | **54.27** |
| | Qwen2.5-72B-Instruct | Evol-Instruct | 40.24 | 42.07 | 43.29 | 46.34 |
| | | Persona Hub | 41.46 | 42.07 | 45.12 | 46.95 |
| | | Temp. Sampling | 39.63 | 42.68 | 45.73 | 46.34 |
| | | TREESYNTH | **46.34** | **48.17** | **50.61** | **53.05** |
| Qwen2.5-7B | Code Alpaca Python subset | Vanilla Data | - | 77.44 | - | - |
| | GPT-4o | Evol-Instruct | 78.05 | 80.49 | 80.49 | 81.10 |
| | | Persona Hub | 76.22 | 77.44 | 78.66 | 79.88 |
| | | Temp. Sampling | 79.27 | 80.49 | 81.10 | 81.71 |
| | | TREESYNTH | **79.88** | **80.49** | **81.49** | **83.05** |
| | LLaMA3.3-70B-Instruct | Evol-Instruct | 75.61 | 76.83 | 77.44 | 78.05 |
| | | Persona Hub | 75.61 | 75.00 | 76.22 | 78.05 |
| | | Temp. Sampling | 76.22 | 76.83 | 77.44 | 77.44 |
| | | TREESYNTH | **77.44** | **78.05** | **79.88** | **81.71** |
| | Qwen2.5-72B-Instruct | Evol-Instruct | 76.22 | 76.22 | 78.66 | 81.71 |
| | | Persona Hub | 76.22 | 76.22 | 76.83 | 77.44 |
| | | Temp. Sampling | 75.00 | 78.05 | 78.66 | 80.49 |
| | | TREESYNTH | **77.44** | **78.66** | **79.88** | **82.32** |

Table 8: Comparison of instruction-tuned model performance on HumanEval using training data from different sources and generation methods. For each data scale (1k, 2k, 10k, 100k), models are fine-tuned on equally sized datasets constructed via various data synthesis methods, as well as the common training set for coding task. "Temp. Sampling" is abbreviated from "Temperature Sampling". "Vanilla Data" denotes the Code Alpaca training sets.

| Foundation Model | Benchmark | Method | | | | | | |
|---|---|---|---|---|---|---|---|---|
| | | Zero-Shot | Few-Shot | Vanilla Data | Temp. Sampling | Evol-Instruct | Persona Hub | TREESYNTH |
| Gemma-3-PT-4B | **MBPP** | 29.80 | - | 41.20 | 42.20 | 42.20 | 42.80 | **44.20** |
| | **HumanEval** | 33.50 | - | 34.76 | 35.98 | 37.20 | 37.80 | **42.07** |
| | **MATH** | 5.20 | 23.80 | 20.00 | 25.30 | 26.60 | 27.20 | **30.30** |
| | **Avg.** | 22.83 | - | 31.99 | 34.49 | 35.33 | 35.93 | **38.86** |
| Gemma-3-PT-12B | **MBPP** | 54.40 | - | 56.60 | 57.00 | 56.60 | 56.20 | **57.40** |
| | **HumanEval** | 59.10 | - | 56.10 | 54.27 | 57.32 | 56.71 | **62.80** |
| | **MATH** | 23.90 | 31.30 | 34.90 | 47.30 | 49.20 | 50.10 | **54.20** |
| | **Avg.** | 45.80 | - | 49.20 | 52.86 | 54.37 | 54.34 | **58.13** |

Table 9: Model performance comparison of various methods with GPT-4o-powered data synthesis across Gemma-3-PT-4B/12B models and multiple benchmarks. "Zero-Shot" and "Few-Shot" exhibit the base performance of foundation models. "Temp. Sampling" is abbreviated from "Temperature Sampling". "Vanilla Data" denotes the original MATH training set for MATH benchmark, and the Code Alpaca Python subset for HumanEval and MBPP. Bold and underlining indicate the best and second-best indicators, respectively. "Avg." means the average of the performance scores across all the benchmarks.

| Model | Benchmark | 1K | 2K | 7.5K | 10K | 100K |
|---|---|---|---|---|---|---|
| | **MATH** | 28.5 | – | 30.3 | 30.7 | 35.2 |
| `Gemma-3-PT-4B` | **MBPP** | 42.4 | 44.2 | – | 45.4 | 47.8 |
| | **HumanEval** | 39.0 | 42.1 | – | 43.9 | 45.7 |
| | **MATH** | 53.2 | – | 54.2 | 55.6 | 57.3 |
| `Gemma-3-PT-12B` | **MBPP** | 55.2 | 57.4 | – | 58.8 | 60.2 |
| | **HumanEval** | 61.6 | 62.8 | – | 63.4 | 66.5 |

Table 10: Evaluation of TREESYNTH's scalability performance across varying data volumes (1K, 2K, 7.5K, 10K, and 100K samples) on `Gemma-3-PT-4B` and `Gemma-3-PT-12B` foundation models with data synthesis powered by `GPT-4o`. Results demonstrate that TREESYNTH maintains strong and consistent scaling behavior across all benchmarks (MATH, MBPP, HumanEval), with downstream task performance steadily improving as synthetic dataset size increases, validating the method's robust scalability properties.

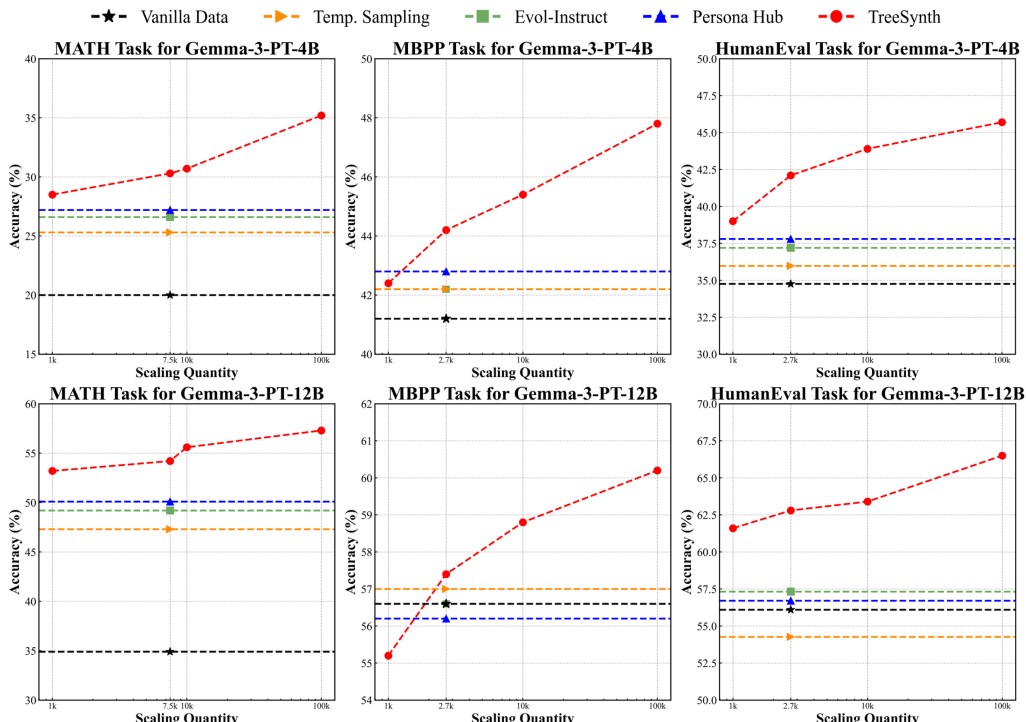

Figure 9: Model performance trends across data scales for different methods powered by `GPT-4o`. "Temp. Sampling" refers to Temperature Sampling. "Vanilla Data" denotes original GSM8K and MATH training sets, and Code Alpaca Python subset for HumanEval and MBPP.

The total compute budget amounted to roughly 30,000 GPU-hours. The software stack comprised PyTorch 2.6.0 linked against CUDA 12.1 (NCCL 2.17.1).

## A.9 Broader Impacts

TREESYNTH promotes AI advancement by autonomously generating diverse and balanced datasets, reducing dependence on costly human curation and mitigating biases imposed by models, seed data and low-variation prompts. This is important for LLM customization and further enhancing their specific capabilities, especially on domains without source data.

| Dataset | Input Tokens | Output Tokens | Input Cost (USD) | Output Cost (USD) | Total (USD) |
|---------|-------------|---------------|-------------------|--------------------|-------------|
| GSM8K Style | 23.1M | 65.4M | 0.88 | 7.84 | 8.72 |
| MATH Style | 7.0M | 25.6M | 0.26 | 3.07 | 3.33 |
| Code Alpaca Style | 25.1M | 56.1M | 0.95 | 6.73 | 7.68 |

Table 11: Detailed breakdown of estimated token consumption (input and output) and associated API costs for synthesizing 100K-scale GSM8K-style, MATH-style, and Code Alpaca-style datasets using `LLaMA3.1-70B-Instruct` through the TREESYNTH framework, demonstrating the cost-effectiveness of the approach.

| Dataset | Metric | Root | Level 1 | Level 2 | Level 3 | Level 4 | Total |
|---------|--------|------|---------|---------|---------|---------|-------|
| Code Alpaca | Number of Criteria | 1 | 12 | 247 | 3853 | – | 4113 |
|  | Number of Attribute Values | 1 | 12 | 247 | 3853 | 57683 | 61796 |
| GSM8K | Number of Criteria | 1 | 13 | 284 | 3914 | – | 4212 |
|  | Number of Attribute Values | 1 | 13 | 284 | 3914 | 52721 | 56933 |
| MATH | Number of Criteria | 1 | 7 | 113 | 1246 | – | 1367 |
|  | Number of Attribute Values | 1 | 7 | 113 | 1246 | 15794 | 17161 |

Table 12: Comprehensive statistics showing the exponential growth of criteria and attribute values across different tree depth levels (Root to Level 4) for Code Alpaca, GSM8K, and MATH datasets, demonstrating how TREESYNTH's spatial partitioning tree progressively refines the data space with increasingly fine-grained descriptors at deeper levels (maximum tree depth set to 4).

```
Root(MATH) [Mathematical Domain]
    |-- Algebra [problem_type]
        |-- equation_solving [computation_complexity]
            |-- low [equation_structure]
                |-- simple_polynomial
                |-- exponential
                |-- ...
            |-- medium [Equation Type]
                |-- Quadratic
                |-- Cubic
                |-- ...
            |-- ...
        |-- ...
    |-- Probability [Problem Complexity]
        |-- ...
    |-- ...
```

Figure 10: Illustrative example of a partial subtree extracted from the MATH spatial partitioning tree, showing the hierarchical structure from the root node through multiple levels of refinement. Each level is annotated with the partition criterion in brackets (*e.g.*, Mathematical Domain, problem_type, computation_complexity), demonstrating how TREESYNTH systematically organizes the mathematical problem space into increasingly specific subspaces.

**Example: Number Theory**
**Question**: Compute $(3^{-1} + 5^{-1})^{-1} \mod 31$. Express your answer as an integer from 0 to 30, inclusive.
**Correct Answer**: 29
**TREESYNTH Solution**: 29
**TREESYNTH Reasoning**:

1. Find modular inverses: $3^{-1} \equiv 21 \mod 31$ and $5^{-1} \equiv 25 \mod 31$

2. Add the inverses: $21 + 25 = 46 \equiv 15 \mod 31$

3. Find inverse of 15: $15^{-1} \equiv 29 \mod 31$ (since $15 \times 29 = 435 \equiv 1 \mod 31$)

1. Find modular inverses: $3^{-1} \equiv 21 \mod 31$ and $5^{-1} \equiv 25 \mod 31$

2. Add the inverses: $21 + 25 = 46 \equiv 15 \mod 31$

3. **ERROR**: Claims $15^{-1} \equiv 21 \mod 31$, but $15 \times 21 = 315 \equiv 5 \mod 31 \neq 1$

The final answer is: 21

Figure 11: Qualitative case study from the Number Theory domain comparing the reasoning quality of models trained on TREESYNTH versus Persona Hub synthetic data when solving a modular inverse problem. The example demonstrates that the TREESYNTH-trained model correctly computes the modular inverse at each step and arrives at the correct answer (29), while the Persona Hub-trained model makes a critical error in the final inverse calculation, yielding an incorrect answer (21). This highlights how TREESYNTH's superior data diversity enables more robust reasoning in specialized mathematical domains.

```
{
  "criterion": "geometric_object_type",
  "attributes": {
    "triangle": [
      "A triangle has sides of lengths 7, 24, and 25.
       Determine the area of the triangle."
    ],
    "regular_polygon": [
      "In a regular hexagon with side length s = 6,
       calculate the area of the hexagon.",
      "A regular dodecagon (12-sided polygon) is inscribed
       in a circle of radius R = 6. Compute the area of the dodecagon."
    ],
    "cyclic_quadrilateral": [
      "A cyclic quadrilateral has sides of lengths 7, 8, 9, and 10.
       Calculate the area of the quadrilateral using Brahmagupta's formula."
    ],
    "composite_figure": [
      "A circle is inscribed in a square, and the square has side length 10.
       Compute the area of the region outside the circle but inside the square."
    ]
  }
}
```

Figure 12: Example output demonstrating how the LLM automatically identifies an appropriate partitioning criterion (`geometric_object_type`) and corresponding attribute values (triangle, regular_polygon, cyclic_quadrilateral, composite_figure) when given five diverse MATH-style geometry problems as pivot samples. This illustrates TREESYNTH's criterion determination mechanism, where the LLM selects dimensions that effectively distinguish between different types of problems, enabling systematic data space partitioning.

| Domain | Persona Hub | TREESYNTH | Improvement over Persona Hub |
|---|---|---|---|
| Number Theory | 58.7% | 71.1% | +12.4% |
| Geometry | 48.9% | 57.8% | +9.0% |
| Counting&Probability | 57.0% | 65.2% | +8.2% |
| Intermediate Algebra | 36.4% | 44.2% | +7.8% |
| Prealgebra | 79.1% | 84.6% | +5.5% |
| Algebra | 83.7% | 88.0% | +4.3% |
| Precalculus | 39.2% | 42.7% | +3.5% |

Table 13: Detailed comparison of accuracy scores across all seven mathematical sub-domains in the MATH benchmark after fine-tuning `Qwen2.5-7B` on synthetic data generated by TREESYNTH versus Persona Hub, showing the absolute performance improvement achieved by TREESYNTH in each domain. Notably, TREESYNTH achieves the most significant gains in specialized domains requiring precise reasoning such as Number Theory (+12.4%), Geometry (+9.0%), and Counting&Probability (+8.2%).

**Prompt Template of GSM8K-style Instruction Generation in Temperature Sampling.**

As a math expert, you are tasked to generate 10 GSM8K-style math word problems suitable for a bright middle school student.

Each question should meet the following criteria:
1. Format: Write problems as real-world word problems that require mathematical reasoning to solve.
2. Step Count: Require between 2 and 8 steps to solve.
3. Operations: Utilize basic arithmetic operations: addition (+), subtraction (-), multiplication (*), and division (/).
4. Complexity: Vary in context and complexity, but REMAIN ACCESSIBLE TO MIDDLE SCHOOL STUDENTS!
5. Clarity: Provide clear, concise questions that encourage step-by-step calculations to reach the final answer.
6. Language: Use natural, conversational language to describe situations while keeping problems clear and unambiguous.
7. Diversity: Ensure that the questions are diverse and distinct from one another from all potential perspectives.

Organize your responses in the following format without any extra text or explanations:
Question 1: text
Question 2: text
...
Question 10: text

Figure 13: Prompt template of GSM8K-style instruction generation in Temperature Sampling.

**Prompt Template of MATH-style Instruction Generation in Temperature Sampling.**

As a math expert, generate 10 high school competition-level math questions in MATH dataset style.

Each question should strictly adhere to these criteria:
1. Question Type: Questions must exclusively involve advanced mathematical reasoning suitable for competitions such as AMC 8, AMC 10, AMC 12, AIME, or USAMO.
2. Difficulty Levels: Clearly assign each question a difficulty level from 1 (easy, typical of early AMC 8 questions) to 5 (very challenging, similar to later AIME/USAMO questions), consistent with recognized mathematical competition standards.
3. Verb and Phrasing Diversity: Employ varied verbs and diverse phrasing, blending both clear interrogative questions and direct imperative instructions to maintain instruction diversity.
4. Clarity and Uniqueness: Questions must provide all necessary details for a solver to determine exactly one unique solution without ambiguity.
5. Notation and Formatting: Use clear and precise mathematical notation written in LATEX. If diagrams or illustrations are necessary, describe them explicitly in descriptive text or represent them using valid Asymptote code.
6. Solvability: Questions should be solvable by advanced high school-level mathematical reasoning without calculators or external computational resources.

Organize your responses strictly in the following format without additional text or explanations:
<Question 1>
  [Question text with proper mathematical notation]
<Difficulty>
  [1-5]

<Question 2>
  [Question text with proper mathematical notation]
<Difficulty>
  [1-5]

...

<Question 10>
  [Question text with proper mathematical notation]
<Difficulty>
  [1-5]

Figure 14: Prompt Template of MATH-style Instruction Generation in Temperature Sampling.

**Prompt Template of Code Alpaca-style Instruction Generation in Temperature Sampling.**

As a coding expert, you are asked to come up with 10 diverse Code Alpaca-style python code generation instruction-input pairs.
Each instruction should meet the following criteria:
1. Instruction type: generated instructions can only be code generation tasks.
2. Verb Variation: Avoid repeating the same verbs across instructions to maintain variety.
3. Phrasing Diversity: Incorporate diverse phrasing, blending questions and commands.
4. Task Variety: Provide different Python programming tasks, e.g. open-ended generation, classification, editing, optimization, etc.
5. Solvable Requests: Ensure the instruction-input pair is solvable by GPT alone, e.g. avoid tasks requiring multimedia or file inputs, etc.
6. Restriction: The code generated by the instructions does not require access to any external resources, including applications, files, systems, or networks. It should be executed solely in the Python console.
7. English Composition: Present instructions in English.
8. Length Restriction: Limit each instruction to one or two sentences, either imperative or interrogative.
9. Input Specificity: When needed, offer a realistic input under 100 words that is detailed enough to evaluate the instruction.
10. Realistic Data: The input should involve realistic data and should not contain simple placeholders.
11. Input constraints: The input must be a common data type in Python and cannot be a python function.
12. Programming Focus: All instructions must relate to coding or programming.
Organize your responses in the following format without any extra text or explanations:
<Instruction 1>
    text of Instruction 1
<Input>
    text Input 1

<Instruction 2>
    text of Instruction 2
<Input>
    text Input 2

...

<Instruction 10>
    text of Instruction 10
<Input>
    text Input 10

Figure 15: Prompt template of Code Alpaca-style instruction generation in Temperature Sampling.

**Prompt Template of SimpleToM-style Instruction Generation in Temperature Sampling.**

Your task is to create <batch_size> short stories where information asymmetry naturally exists. The goal is to generate training data that helps an LLM anticipate behavior based on mental states rather than observable clues. Below are some instructions to follow for each instance.
1. Write a two-sentence story.
   - Decide how to instantiate the main entities in the story:
      - Person X (required): either a real, creative name followed by a simple descriptor indicating their role in the story or a group of people.
      - Object Z / Person Z / Action Z (required): This will be the subject of the KEY INFORMATION.
      - Person Y (optional): any additional character or group if needed for the story, but is not required.
   - For the FIRST SENTENCE of the story, write the KEY INFORMATION about Object Z / Person Z / Action Z (and Person Y) that is unknown to Person X (due to the general reason given in the scenario). Person X should not be able to observe this KEY INFORMATION through their actions in the story (either implicit or explicit actions). DO NOT use information which might be observed by Person X through normal, careful observation (such as "expiration date", "leaking container", "smell", etc).
   - The SECOND SENTENCE of the story is about what Person X will usually do regarding Object Z / Person Z / Action Z (and Person Y) in the scenario (ignoring the KEY INFORMATION). This sentence should describe what the character does using fine-grained actions. DO NOT include any descriptions which involve the emotions or thoughts of Person X, just describe actions.
2. Generate Question & Answer Choices:
   - Write a question predicting what Person X will likely do next.
   - Provide two verbal action choices:
      - (A): Correct action to the question (given the fact that person X is not aware of the KEY INFORMATION). Make sure the story does not have any mention of this action.
      - (B): Counterfactual (incorrect) action to the question. This answer should be a likely answer to the question under the assumption that person X somehow has full access to the KEY INFORMATION after all (maybe only possible using "magic" or some omnipotent skill).
   - Each action should be a complete but concise verbal phrase, without adjectives or adverbs. Avoid making it too short or too detailed.
   - Ensure two choices are in the **same length**.
3. Give the Final Answer:
   - Provide a short chain-of-thought explaining why the correct answer is (A).
Now, organize your response (<batch_size> instances) in the following format. Separate each instance using **only a blank line** (no extra dividers or explanations).
Instance <N>:
[INPUT]
Given the following story, answer the question by giving the correct answer choice, (A) or (B).
Story: <the two-sentence story>
Question: <the question>
(A) <action choice when X is unaware of the key information>
(B) <choice when X has full knowledge of the key information>
What is the correct answer?

[ANSWER]
<the chain-of-thought>. So the answer is (A).

Figure 16: Prompt template of SimpleToM-style instruction generation in Temperature Sampling.

---

**Prompt Template of GSM8K-style Instruction Generation in TREESYNTH.**

As a math expert, you are tasked to generate 10 GSM8K-style math word problems suitable for a bright middle school student.

Each question should meet the following criteria:
1. Format: Write problems as real-world word problems that require mathematical reasoning to solve.
2. Step Count: Require between 2 and 8 steps to solve.
3. Operations: Utilize basic arithmetic operations: addition (+), subtraction (-), multiplication (*), and division (/).
4. Complexity: Vary in context and complexity, but REMAIN ACCESSIBLE TO MIDDLE SCHOOL STUDENTS!
5. Clarity: Provide clear, concise questions that encourage step-by-step calculations to reach the final answer.
6. Language: Use natural, conversational language to describe situations while keeping problems clear and unambiguous.
7. Diversity: Ensure that the questions are diverse and distinct from one another from all potential perspectives.
8. Attributes: Each problem should be associated with all these attributes: <Attributes>

Organize your responses in the following format without any extra text or explanations:
Question 1: text
Question 2: text
...
Question 10: text

Figure 17: Prompt template of GSM8K-style instruction generation in TREESYNTH.

**Prompt Template of MATH-style Instruction Generation in TREESYNTH.**

As a math expert, generate 10 high school competition-level math questions in MATH dataset style.

Each question should strictly adhere to these criteria:
1. Question Type: Questions must exclusively involve advanced mathematical reasoning suitable for competitions such as AMC 8, AMC 10, AMC 12, AIME, or USAMO.
2. Difficulty Levels: Clearly assign each question a difficulty level from 1 (easy, typical of early AMC 8 questions) to 5 (very challenging, similar to later AIME/USAMO questions), consistent with recognized mathematical competition standards.
3. Verb and Phrasing Diversity: Employ varied verbs and diverse phrasing, blending both clear interrogative questions and direct imperative instructions to maintain instruction diversity.
4. Clarity and Uniqueness: Questions must provide all necessary details for a solver to determine exactly one unique solution without ambiguity.
5. Notation and Formatting: Use clear and precise mathematical notation written in LATEX. If diagrams or illustrations are necessary, describe them explicitly in descriptive text or represent them using valid Asymptote code.
6. Solvability: Questions should be solvable by advanced high school-level mathematical reasoning without calculators or external computational resources.
7.Attributes: Each question should be associated with all these attributes: <Attributes>

Organize your responses strictly in the following format without additional text or explanations:
<Question 1>
    [Question text with proper mathematical notation]
<Difficulty>
    [1-5]

<Question 2>
    [Question text with proper mathematical notation]
<Difficulty>
    [1-5]

...

<Question 10>
    [Question text with proper mathematical notation]
<Difficulty>
    [1-5]

Figure 18: Prompt template of MATH-style instruction generation in TREESYNTH.

**Prompt Template of Code Alpaca-style Instruction Generation in TREESYNTH.**

As a coding expert, you are asked to come up with 10 diverse Code Alpaca-style python code generation instruction-input pairs.
Each instruction should meet the following criteria:
1. Instruction type: generated instructions can only be code generation tasks.
2. Verb Variation: Avoid repeating the same verbs across instructions to maintain variety.
3. Phrasing Diversity: Incorporate diverse phrasing, blending questions and commands.
4. Task Variety: Provide different Python programming tasks, e.g. open-ended generation, classification, editing, optimization, etc.
5. Solvable Requests: Ensure the instruction-input pair is solvable by GPT alone, e.g. avoid tasks requiring multimedia or file inputs, etc.
6. Restriction: The code generated by the instructions does not require access to any external resources, including applications, files, systems, or networks. It should be executed solely in the Python console.
7. English Composition: Present instructions in English.
8. Length Restriction: Limit each instruction to one or two sentences, either imperative or interrogative.
9. Input Specificity: When needed, offer a realistic input under 100 words that is detailed enough to evaluate the instruction.
10. Realistic Data: The input should involve realistic data and should not contain simple placeholders.
11. Input constraints: The input must be a common data type in Python and cannot be a python function.
12. Programming Focus: All instructions must relate to coding or programming.
13.Attributes: Each problem should be associated with all these attributes: <Attributes>
Organize your responses in the following format without any extra text or explanations:
<Instruction 1>
    text of Instruction 1
<Input>
    text Input 1

<Instruction 2>
    text of Instruction 2
<Input>
    text Input 2

...

<Instruction 10>
    text of Instruction 10
<Input>
    text Input 10

Figure 19: Prompt template of Code Alpaca-style instruction generation in TREESYNTH.

**Prompt Template of SimpleToM-style Instruction Generation in TREESYNTH.**

Your task is to create <n_samples> short stories where information asymmetry naturally exists. The goal is to generate training data that helps an LLM anticipate behavior based on mental states rather than observable clues. Below are some instructions to follow for each instance.
1. Write a two-sentence story.
  - Decide how to instantiate the main entities in the story:
    - Person X (required): either a real, creative name followed by a simple descriptor indicating their role in the story or a group of people.
    - Object Z / Person Z / Action Z (required): This will be the subject of the KEY INFORMATION.
    - Person Y (optional): any additional character or group if needed for the story, but is not required.
  - For the FIRST SENTENCE of the story, write the KEY INFORMATION about Object Z / Person Z / Action Z (and Person Y) that is unknown to Person X (due to the general reason given in the scenario). Person X should not be able to observe this KEY INFORMATION through their actions in the story (either implicit or explicit actions). DO NOT use information which might be observed by Person X through normal, careful observation (such as "expiration date", "leaking container", "smell", etc).
  - The SECOND SENTENCE of the story is about what Person X will usually do regarding Object Z / Person Z / Action Z (and Person Y) in the scenario (ignoring the KEY INFORMATION). This sentence should describe what the character does using fine-grained actions. DO NOT include any descriptions which involve the emotions or thoughts of Person X, just describe actions.
2. Generate Question & Answer Choices:
  - Write a question predicting what Person X will likely do next.
  - Provide two verbal action choices:
    - (A): Correct action to the question (given the fact that person X is not aware of the KEY INFORMATION). Make sure the story does not have any mention of this action.
    - (B): Counterfactual (incorrect) action to the question. This answer should be a likely answer to the question under the assumption that person X somehow has full access to the KEY INFORMATION after all (maybe only possible using "magic" or some omnipotent skill).
    - Each action should be a complete but concise verbal phrase, without adjectives or adverbs. Avoid making it too short or too detailed.
    - Ensure two choices are in the **same length**.
3. Give the Final Answer:
  - Provide a short chain-of-thought explaining why the correct answer is (A).

### All stories MUST be **accociated with the ATTRIBUTES below**:
<attributes_json>
Now, organize your response (<n_samples> instances) in the following format. Separate each instance using **only a blank line** (no extra dividers or explanations).
Instance <N>:
[INPUT]
Given the following story, answer the question by giving the correct answer choice, (A) or (B).
Story: <the two-sentence story>
Question: <the question>
(A) <action choice when X is unaware of the key information>
(B) <choice when X has full knowledge of the key information>
What is the correct answer?

[ANSWER]
<the chain-of-thought>. So the answer is (A).

Figure 20: Prompt template of SimpleToM-style instruction generation in TREESYNTH.

**Prompt Template of GSM8K-style Instruction Criterion Determination in TREESYNTH.**

As an analysis expert, your task is to examine the following questions to identify the SINGLE most significant dimension that characterizes the question space and differentiates these questions.
Questions:
<Samples>

Dimension Requirements:
1. Core Dimension Identification: Identify exactly ONE core dimension that best distinguishes these questions.
2. Excluded Dimensions: <Dimensions>
3. Unique Categorization: Each question MUST be categorized into exactly ONE attribute value.
4. Mutually Exclusive Values: Attribute values must be mutually exclusive.
5. Clarity in Values: Avoid ambiguous attribute values, such as "others".
6. Independent Values: Each attribute must be a single distinct value - NO combined values like "attribute1_and_attribute2" or "attribute1/attribute2"! Each attribute must be a single distinct value - NO combined values like "attribute1_and_attribute2" or "attribute1/attribute2"! Each attribute must be a single distinct value - NO combined values like "attribute1_and_attribute2" or "attribute1/attribute2"!

Organize your responses in the following format without any extra text or explanations:
{{
"dimension": "dimension_name",
"attributes": {{
   "attribute1": [list of sample indices],
   "attribute2": [list of sample indices],
   ...
}}
}}

Figure 21: Prompt template of GSM8K-style instruction criterion determination in TREESYNTH.

---

**Prompt Template of MATH-style Instruction Criterion Determination in TREESYNTH.**

As a math expert, your task is to examine the following MATH questions identify the SINGLE most significant dimension that characterizes the question space and differentiates these questions.
Questions:
<Samples>

Dimension Requirements:
1. Core Dimension Identification: Identify exactly ONE core dimension that best distinguishes these questions.
2. Excluded Dimensions: <Dimensions>
3. Unique Categorization: Each question MUST be categorized into exactly ONE attribute value.
4. Mutually Exclusive Values: Attribute values must be mutually exclusive.
5. Clarity in Values: Avoid ambiguous attribute values, such as "others".
6. Independent Values: Each attribute must be a single distinct value - NO combined values like "attribute1_and_attribute2" or "attribute1/attribute2".

Organize your responses in the following format without any extra text or explanations:
{{
"dimension": "dimension_name",
"attributes": {{
   "attribute1": [list of sample indices],
   "attribute2": [list of sample indices],
   ...
}}
}}

Figure 22: Prompt template of MATH-style instruction criterion determination in TREESYNTH.

**Prompt Template of Code Alpaca-style Instruction Criterion Determination in TREESYNTH.**

As a coding expert, your task is to examine the following python code generation instruction-input pairs to identify the SINGLE most significant dimension that characterizes the instruction-input space and differentiates these instruction-input pairs.
Instruction-input pairs:
<Samples>

Dimension Requirements:
1. Core Dimension Identification: Identify exactly ONE core dimension that best distinguishes these instruction-input pairs.
2. Excluded Dimensions: <Dimensions>
3. Unique Categorization: Each instruction-input pair MUST be categorized into exactly ONE attribute value.
4. Mutually Exclusive Values: Attribute values must be mutually exclusive.
5. Clarity in Values: Avoid ambiguous attribute values, such as "others".
6. Independent Values: Each attribute must be a single distinct value - NO combined values like "attribute1_and_attribute2" or "attribute1/attribute2".

Organize your responses in the following format without any extra text or explanations:
{{
"dimension": "dimension_name",
"attributes": {{
    "attribute1": [list of sample indices],
    "attribute2": [list of sample indices],
    ...
}}
}}

Figure 23: Prompt template of Code Alpaca-style instruction criterion determination in TREESYNTH.

**Prompt Template of SimpleToM-style Instruction Criterion Determination in TREESYNTH.**

Below are some stories that take place in real-world scenarios where unawareness and information asymmetry with various underlying reasons naturally exists. As an expert equipped with rich commonsense and extensive knowledge, your task is to examine the following stories to identify the SINGLE most significant dimension that characterizes the story space and differentiates these stories.

Stories:
<Samples>

Dimension Requirements:
1. Core Dimension Identification: Identify exactly ONE core dimension that best distinguishes these stories.
2. Excluded Dimensions: <Dimensions>
3. Unique Categorization: Each question MUST be categorized into exactly ONE attribute value.
4. Mutually Exclusive Values: Attribute values must be mutually exclusive.
5. Clarity in Values: Avoid ambiguous attribute values, such as "others".
6. Independent Values: Each attribute must be a single distinct value - NO combined values like "attribute1_and_attribute2" or "attribute1/attribute2"!

Organize your responses in the following format without any extra text or explanations:
{{
"dimension": "dimension_name",
"attributes": {{
    "attribute1": [list of sample indices],
    "attribute2": [list of sample indices],
    ...
}}
}}

Figure 24: Prompt template of SimpleToM-style instruction criterion determination in TREESYNTH.

**Prompt Template of GSM8K-style Instruction Subspace Coverage in TREESYNTH.**

As an analysis expert, your task is to supplement the potential attribute values for a specified dimension in order to comprehensively model the entire space of questions.

Dimension: <Dimension>
Exiting attributes values: <Attribute values>

Requirements for New Attribute Values:
1. Clarity: Avoid ambiguous values, such as "others".
2. Mutual Exclusivity: Ensure that attribute values do not overlap.
3. Completeness: Ensure that all possible attribute values fully cover the dimension.
4. GRADE LEVEL: Keep all values within elementary and middle school students' understanding! Keep all values within elementary and middle school students' understanding! Keep all values within elementary and middle school students' understanding!
5. SIMPLICITY: Use basic, straightforward terms that young students can understand! Use basic, straightforward terms that young students can understand! Use basic, straightforward terms that young students can understand!

Organize your responses in the following format without any extra text or explanations:
- If the existing attribute values completely cover the entire dimension, only output "null". For example,
null
- If the number of potential attribute values is more than 10, first output 10 potential new attribute values, and end your output with "infinite" in a new line. For example,
attribute value 1
attribute value 2
...
attribute value 10
infinite
- Otherwise, output all the potential new attribute values, and end your output with "complete" in a new line. For example,
attribute value 1
attribute value 2
...
attribute value n
complete

Figure 25: Prompt template of GSM8K-style instruction subspace coverage in TREESYNTH.

**Prompt Template of MATH-style Instruction Subspace Coverage in TREESYNTH.**

As a math expert, your task is to supplement the potential attribute values for a specified dimension in order to comprehensively model the entire space of questions.
The whole space of questions is described as follows:
1. Question Type: Questions must exclusively involve advanced mathematical reasoning suitable for competitions such as AMC 8, AMC 10, AMC 12, AIME, or USAMO.
2. Difficulty Levels: Clearly assign each question a difficulty level from 1 (easy, typical of early AMC 8 questions) to 5 (very challenging, similar to later AIME/USAMO questions), consistent with recognized mathematical competition standards.
3. Verb and Phrasing Diversity: Employ varied verbs and diverse phrasing, blending both clear interrogative questions and direct imperative instructions to maintain instruction diversity.
4. Clarity and Uniqueness: Questions must provide all necessary details for a solver to determine exactly one unique solution without ambiguity.
5. Notation and Formatting: Use clear and precise mathematical notation written in LATEX. If diagrams or illustrations are necessary, describe them explicitly in descriptive text or represent them using valid Asymptote code.
6. Solvability: Questions should be solvable by advanced high school-level mathematical reasoning without calculators or external computational resources.
Dimension: <Dimension>
Exiting attributes values: <Attribute values>
Requirements for New Attribute Values:
1. Clarity: Avoid ambiguous values, such as "others".
2. Mutual Exclusivity: Ensure that attribute values do not overlap.
3. Completeness: Ensure that all possible attribute values fully cover the dimension.
4. Mathematical Complexity: Generate attribute values that reflect high school competition-level mathematical techniques and concepts.
Organize your responses in the following format without any extra text or explanations:
- If the existing attribute values completely cover the entire dimension, only output "null". For example,
null
- If the number of potential attribute values is more than 10, first output 10 potential new attribute values, and end your output with "infinite" in a new line. For example,
attribute value 1
attribute value 2
...
attribute value 10
infinite
- Otherwise, output all the potential new attribute values, and end your output with "complete" in a new line. For example,
attribute value 1
attribute value 2
...
attribute value n
complete

Figure 26: Prompt template of MATH-style instruction subspace coverage in TREESYNTH.

**Prompt Template of Code Alpaca-style Instruction Subspace Coverage in TREESYNTH.**

As a coding expert, your task is to supplement the potential attribute values for a specified dimension in order to comprehensively model the entire space of instruction-input pairs.

The whole space of instruction-input pairs is described as follows:

1. Instruction type: generated instructions can only be code generation tasks.

2. Various Python programming tasks are included, e.g. open-ended generation, classification, editing, optimization, etc.

3. Each instruction-input pair is solvable by GPT alone, e.g. avoid tasks requiring multimedia or file inputs, etc.

4. All the instruction-input pairs are in English.

5. When unnecessary, the input may be omitted in the instruction-input pairs.

6. Restriction: The code generated by the instructions does not require access to any external resources, including applications, files, systems, or networks. It should be executed solely in the Python console.

Dimension: <Dimension>

Exiting attributes values: <Attribute values>

Requirements for New Attribute Values:

1. Clarity: Avoid ambiguous values, such as "others".

2. Mutual Exclusivity: Ensure that attribute values do not overlap.

3. Completeness: Ensure that all possible attribute values fully cover the dimension.

4. Attribute Scope: All attributes should be python coding or programming related.

Organize your responses in the following format without any extra text or explanations:

- If the existing attribute values completely cover the entire dimension, only output "null". For example,

null

- If the number of potential attribute values is more than 10, first output 10 potential new attribute values, and end your output with "infinite" in a new line. For example,

attribute value 1

attribute value 2

...

attribute value 10

infinite

- Otherwise, output all the potential new attribute values, and end your output with "complete" in a new line. For example,

attribute value 1

attribute value 2

...

attribute value n

complete

Figure 27: Prompt template of Code Alpaca-style instruction subspace coverage in TREESYNTH.

**Prompt Template of SimpleToM-style Instruction Subspace Coverage in TREESYNTH.**

As an analysis expert, your task is to supplement the potential attribute values for a specified dimension in order to comprehensively model the entire space of stories. Note that these stories take place in real-world scenarios where information asymmetry naturally exists, with various underlying causes.

Dimension: <Dimension>
Existing attributes values: <Attribute values>

Requirements for New Attribute Values:
1. Clarity: Avoid ambiguous values, such as "others".
2. Mutual Exclusivity: Ensure that attribute values do not overlap with each other or with the existing values.
3. Completeness: Ensure that all possible attribute values fully cover the dimension.
4. Harmfulness and Unethicality: Avoid

Organize your responses in the following format without any extra text or explanations:
- If the existing attribute values completely cover the entire dimension, only output "null". For example,
null
- If the number of potential attribute values is more than 10, first output 10 potential new attribute values, and end your output with "infinite" in a new line. For example,
attribute value 1
attribute value 2
...
attribute value 10
infinite
- Otherwise, output all the potential new attribute values, and end your output with "complete" in a new line. For example,
attribute value 1
attribute value 2
...
attribute value n
complete

Figure 28: Prompt template of SimpleToM-style instruction subspace coverage in TREESYNTH.

**Prompt Template of Converting Negative Samples of SimpleToM-style Data into Positive Samples.**

You are given a short, two-sentence story illustrating information asymmetry, where Person X is unaware of a crucial fact about Y. Your task is to convert the story into an "information symmetry" version by adding a **subtle environmental or background clue** **only in the first sentence**. Importantly, do **not** describe Person X actively noticing or becoming aware of the clue; just insert a slight detail implying something might be wrong or unusual.
**Detailed Instructions**
1. **Original Story** (two sentences):
   - Sentence 1 reveals the hidden fact (which X originally does not know).
   - Sentence 2 describes Person X's action, still unaware.
2. **Modified Story** (two sentences):
   - **Sentence 1**: Insert a minor clue that could lead X (or a reader) to infer the hidden fact **without** explicitly saying "X notices" or "X realizes." The clue should be subtle and not directly point to the hidden fact.
   - **Sentence 2**: Keep it almost the same as in the original story, unless trivial edits are needed for coherence. Avoid stating that X has already changed behavior. The point is that X **could** have inferred the fact from the background detail, but the text does not explicitly say so.
3. **Question & Choices**:
   - Use the same question from the original story or rephrase it slightly as "What does X do next?"
   - (A) The original uninformed action.
   - (B) The new informed action.
   - Each action should be a complete but concise verbal phrase, without adjectives or adverbs. Avoid making it too short or too detailed.
4. **Final Answer**:
   - Provide a short (1–2 sentences) reasoning that references the subtle background clue in the first sentence, leading X to choose (B).
   - End with: "So the answer is (B)."
Here is the original story, question and uninformed action:
Original Story: <story>
Original Question: <ques>
Original Action: <act>
Now, organize your response in the following format. Separate each instance using **only a blank line** (no extra dividers or explanations).
[INPUT]
Given the following story, answer the question by giving the correct answer choice, (A) or (B).
Original Story: <the original two-sentence story given to you which shows the old info asymmetry>
Modified Story: <the new two-sentence story where X has discovered the missing info>
Question: <the question>
(A) <old uninformed action>
(B) <new informed action>
What is the correct answer?
[ANSWER]
<brief chain-of-thought>. So the answer is (B).
### Key Reminders
- Do not say "X notices / sees / realizes / suspects." Instead, simply mention an observable detail in the environment or object. Let the user infer that X **could** realize it.
- Keep the second sentence almost the same.
- The inserted clue must be enough that (B) is justified.
- This ensures the final scenario still requires a bit of inference, rather than the story outright stating X's awareness.

Figure 29: Prompt template of converting negative samples of SimpleToM-style data into positive samples.

---

**Prompt Template of Sample Classification.**

You are tasked with classifying the following story based on the criterion of **<dimension>**.

STORY:
<sample>

You must **select exactly one** category from the option list below that best fits the main characteristics of the story. If none is a perfect fit, select the closest one.

OPTION LIST:
<attributes>

Please strictly follow the output format below and do not output anything else.

Output in JSON format:
{{
   "category": "<selected category, using **exact text** from the list>",
   "explanation": "<one-sentence justification>"
}}

Figure 30: Prompt template of sample classification.

**MATH style data generated by TREESYNTH**

**Attribute value and Dimension:**
1.Attribute value: Algebraic Equations,
   Dimension: Problem Type,
2.Attribute value: Geometry and Measurement,
   Dimension: Mathematical Focus,
3.Attribute value: perimeter/fencing,
   Dimension: object of calculation,
4.Attribute value: Rectangular,
   Dimension: Shape of the Area,

**Generated Samples:**
1."Emma is planning to build a rectangular garden in her backyard, where the length is 3 feet more than twice the width. If she wants the perimeter of the garden to be 30 feet, what are the dimensions of the garden?",
2."A farmer is enclosing a rectangular pasture with a fence. The length is 4 times the width, and the total fencing needed is 60 meters. What are the dimensions of the pasture?",
3."Liam wants to create a rectangular dog park where the length is 5 meters more than the width. If he has 50 meters of fencing to use around the park, what should be the dimensions of the dog park?",
4."Sarah is designing a rectangular flower bed where the length is 7 feet longer than the width. If she has 48 feet of fencing available, what are the dimensions of the flower bed?",
5."A rectangular playground is being designed such that its length is double its width. If the perimeter of the playground is meant to be 72 meters, what are the dimensions of the playground?",
6."A rectangular swimming pool is planned where the length is 10 feet more than the width. If the perimeter needs to be 84 feet, what should be the dimensions of the pool?",
7."David is building a rectangular deck where the length is 6 feet more than the width. If the total perimeter of the deck is to be 40 feet, what are the dimensions of the deck?",
8."A rectangular parking lot is to be constructed with the length being 8 meters more than the width. If the perimeter is supposed to be 100 meters, what will be the dimensions of the parking lot?",
9."A rectangular storage area is being planned with a length that is 3 meters longer than the width. If the entire perimeter is to be 90 meters, what are the dimensions of the storage area?",
10."A rectangular sports field is to be designed where the length is 12 meters more than twice the width. If the perimeter of the field is 96 meters, what are the dimensions of the field?"

Figure 31: GSM8K style data generated by TREESYNTH.

**MATH style data generated by TREESYNTH**

**Attribute value and Dimension:**
1.Attribute value: Algebra,
   Dimension: Mathematical Domain,
2.Attribute value: equation solving,
   Dimension: problem type,
3.Attribute value: low,
   Dimension: computation complexity,
4.Attribute value: simple polynomial,
   Dimension: equation structure,

**Generated Samples:**
1."Solve for (x): $(2x^2 - 5x + 3 = 0)$.  <Difficulty>  1",
2."Determine the real solution to the equation $(x^2 + 4x + 4 = 16)$.  <Difficulty>  1",
3."If $(x^2 + 3x - 10 = 0)$, compute the sum of all real solutions for (x).  <Difficulty>  2",
4."Let $(2x^2 - 3x - 5 = 0)$. Find the product of the roots of the equation.  <Difficulty>  2",
5."Solve for (x): $((x - 2)(x + 3) = 0)$.  <Difficulty>  1",
6."Find the smallest positive solution to the equation $(x^2 - 6x + 8 = 0)$.  <Difficulty>  2",
7."For $(x^2 - 7x + 12 = 0)$, determine the larger root.  <Difficulty>  2",
8."Let $(x^2 + px + 16 = 0)$ have roots 4 and (-4). Find the value of (p).  <Difficulty>  3",
9."Solve for (x): $(3x^2 - 2x - 8 = 0)$. Express your answer in simplest radical form.  <Difficulty>  4",
10."Find all real solutions to the quartic equation $(x^4 - 5x^2 + 4 = 0)$.  <Difficulty>  4"

Figure 32: MATH style data generated by TREESYNTH.

**Code Alpaca style data generated by TREESYNTH**

**Attribute value and Dimension:**
1. Attribute value: numeric computation,
   Dimension: operation type,
2. Attribute value: list of integers,
   Dimension: input data type,
3. Attribute value: central tendency,
   Dimension: required list property or computation,
4. Attribute value: mean,
   Dimension: statistical measure,

**Generated Samples:**
1. "Create a Python function to calculate the mean of a list of integers. <Input>    [10, 20, 30, 40, 50]",
2. "How can you determine the average value of numbers in a list using Python? <Input>    [3, 6, 9, 12, 15]",
3. "Write a script to find the mean of a given list of integers. <Input>    [5, 15, 25, 35, 45]",
4. "What Python code would you use to compute the mean of a list of integers? <Input>    [8, 16, 24, 32, 40]",
5. "Could you draft a Python function that returns the average of numbers in a list? <Input>    [7, 14, 21, 28, 35]",
6. "Generate a Python function that calculates the mean of integers in a list. <Input>    [2, 4, 6, 8, 10]",
7. "Can you write a Python code snippet to find the mean of a list of integers? <Input>    [11, 22, 33, 44, 55]",
8. "Devise a Python script to calculate the average of a list of integers. <Input>    [9, 18, 27, 36, 45]",
9. "What is the Python function to compute the mean of a list of integers? <Input>    [12, 24, 36, 48, 60]",
10. "Formulate a Python solution to find the mean of a list of numbers. <Input>    [1, 3, 5, 7, 9]"

Figure 33: Code Alpaca style data generated by TREESYNTH.

