# OpenReview forum: "TreeSynth: Synthesizing Diverse Data from Scratch via Tree-Guided Subspace Partitioning"
_NeurIPS.cc/2025/Conference — NeurIPS 2025 spotlight_

### Official Review · Reviewer_4JZM · 2025-06-10

**Clarity:** 2
**Significance:** 2
**Originality:** 2
**Rating:** 5
**Confidence:** 4

**Summary:**

This paper proposes TreeSynth, a method to generate data synthetically by formulating concepts into a tree format. Nodes are concepts, where deeper nodes are more fine-grained concepts. They show results on Llama and Qwen 70B models with math and code generation benchmarks, showing improved performance.

**Questions:**

1. How did you choose N to be 10? Were there any hyperparameter studies done?
2. The manuscript mentions "another LLM, proficient in identifying inter-sample distinctions, determines exactly one core criterion" on lines 116-117. It wasn't immediately clear to me what this LLM is - is it fine-tuned, few-shot augmented, or a base model? I might have missed where those details were mentioned.

**Ethical Concerns:**

["NO or VERY MINOR ethics concerns only"]

**Final Justification:**

The authors have addressed my concerns. I also read through the other reviews/responses -- I liked Reviewer W3cd's comment about the computational costs and appreciated the author's detailed analysis.

**Limitations:**

Yes

**Paper Formatting Concerns:**

None, the formatting looks good.

**Quality:**

3

**Strengths And Weaknesses:**

Strengths:
- The paper provides enough mathematical formulation to make it easier to follow
- Experimentation is extensive

Weaknesses:
- This methodology reminds me of the InstructLab paper (LAB: Large-Scale Alignment for ChatBots) that also uses a tree-like data synthesis method. Providing a comparison (results would be nice, but a discussion will suffice for now) would be great.
- I might have missed any human studies or automatic studies to measure the quality of the synthetic data, but that information would also help.

I'd be more than happy to update my scores once my concerns are addressed.

---

> ### Author Rebuttal · Authors · 2025-07-31
>
> Dear Reviewer 4JZM,
>
> Thanks for your dedication to our reviewing process and your comments on our work! We will try our best to clarify your comments one by one below. Hope that we can fully address your concerns!
>
> To begin with, we wanna highlight the difference between criteria and concepts, as well as the significant impact of TreeSynth.
> - **Criteria are more fine-grained than concepts.** For example, while a math problem may involve multiple concepts (e.g., persons, events, locations, mathematical operators), criteria can **additionally contain more compositional and abstract dimensions,** such as the order of mathematical operators and difficulty levels. Furthermore, the **subspace of a leaf node is defined by the intersection of all attribute values** along the path from the root (Sec. 3.2), instead of a single existing concept. This broader coverage and finer granularity allow for a **more precise partitioning of the data space,** thereby enhancing data diversity and model performance.
> - **Higher diversity and better performance:** Data diversity is crucial for preventing data collapse, avoiding overfitting, and enhancing model performance. With the tree-guided complementarity and exhaustiveness, TreeSynth synthesizes diverse and comprehensive domain-specific data, and demonstrates better performance than human-annotated or other synthetic datasets, **offering a superior solution to data scarcity, collapse, and the high cost of manual annotation** (Sec. 4.2).
> - **Excellent Scalability:** Greater scalability implies that model capabilities can continuously improve with more data, ultimately leading to superior performance. TreeSynth exemplifies this by **effectively scaling datasets and consistently improving downstream performance with increased data volume**, showcasing its significant potential for remarkable model performance (Sec. 4.2).
> - **Effective Re-balance of Existing Datasets:** In addition to synthesizing data from scratch, TreeSynth can also be used to re-balance existing datasets, achieving **improved performance** (Sec. 4.3).
>
> **Weakness 1: Comparison with LAB**
> Thank you for your insightful suggestion and bringing paper [1] to our attention! We have thoroughly read LAB, a taxonomy-guided approach for instruction-tuning data synthesis. Due to the unavailability of LAB's dataset and its complete taxonomy, our comparative analysis focuses on its methodology.
> - **LAB necessitates a manually constructed tree-like taxonomy,** which requires further manual modifications when adapted to specific domains. Conversely, **TreeSynth autonomously partitions the data space, and synthesizes a dataset from a task description sentence.**
> - **LAB relies on manually provided seed data,** as mentioned in the paper "the tasks are defined in the leaf nodes and exemplified by providing manually written instruction-response pairs". In contrast, **TreeSynth can generate extensive data from scratch.** This fundamental difference implies that TreeSynth is more accessible and possesses greater potential for widespread applications.
> - As mentioned above, **LAB utilizes a single concept to define a data subspace,** whereas **TreeSynth leverages the intersection of criteria for broader coverage and finer granularity,** which facilitates the generation of a more balanced and comprehensive dataset.
>
> **Weakness 2: Synthetic Data Quality Evaluation**
>
> We assess the quality of synthetic data from three perspectives: data diversity, t-SNE visualization analysis, and downstream task performance.
>
> - **Data diversity** is quantified using cosine similarity [2]. As shown in Tables 1-4, TreeSynth demonstrates significantly superior diversity across all tasks and models compared to human-annotated and other synthetic datasets. Notably, the GPT-4o-driven TreeSynth improves diversity by 12.5%, 25.0%, and 34.5% over the original GSM8K, MATH, and Code Alpaca datasets, respectively.
> - **The t-SNE visualizations** (Figures 4 and 7) illustrate TreeSynth's comprehensive coverage of the data space. By strategically partitioning the space from a global perspective and synthesizing data within these subspaces, TreeSynth overcomes inherent model biases and source data limitations, thereby achieving a more balanced spatial coverage.
> - **Finally, downstream task performance** (Figures 5, 6, 8 and Tables 5-8) directly validates the superiority of our synthetic data. Guided by the global view of its tree structure, **TreeSynth not only outperforms competing methods but also demonstrates excellent scalability,** as downstream performance consistently improves with increasing data volume.
>
> **Question 1: Choice of Hyperparameter N**
> Thanks a lot for your valuable questions! The hyperparameter N sets the threshold for the number of attribute values in a node; nodes exceeding this count are regarded as "infinite nodes". Intuitively, a larger N results in finer partitioning of the data space and better data quality. Based on our preliminary experiments, we chose N=10, which is sufficient to achieve better performance.
>
> **Question 2: Clarification on the Criterion-Determining LLM**
>
> We apologize for the lack of clarity in our previous description!
>
> - Similar to other stages, this "proficient LLM" in Criterion Determination is also an **off-the-shelf LLM** (e.g., LLaMA-70B, Qwen-72B, or GPT-4o), but is instructed to identify the core criterion with the prompt presented in Figures 17-20 (without few shots). This design leverages the powerful capabilities of existing LLMs, while maintaining TreeSynth's ease of use. The excellent performance of TreeSynth validates the efficacy of this approach.
> - As shown below, given 10 MATH-style geometry questions, a "Criterion Determination" LLM can identify "geometric_object_type" as the core criterion to effectively partition the data into four subspaces, before a "Subspace Coverage" LLM complements other potential subspaces.
> ```
> {
> "criterion": "geometric_object_type",
> "attributes": {
>  "triangle": [
>   "A triangle has sides of lengths 7, 24, and 25. Determine the area of the triangle.",
>   "Let ABC be a triangle with AB = 13, BC = 14, and CA = 15. Compute the radius of the circle inscribed in the triangle.",
>   "Let ABC be a triangle where AB = 8, BC = 10, and CA = 12. Find the length of the altitude from A to BC.",
>   "Consider a triangle ABC with AB = 6, BC = 8, and CA = 10. Prove that ABC is a right triangle and then compute the radius of its circumscribed circle.",
>   "In a triangle ABC, the medians from vertices A, B, and C intersect at a point G. If AB = 12, BC = 16, and CA = 20, compute the distance from G to vertex A.",
>   "Let ABC be a triangle where AB = 15, BC = 20, and CA = 25. Determine the coordinates of the orthocenter if vertex A is at (0, 0), B is at (15, 0), and C is at (0, 20)."
>  ],
>  "regular_polygon": [
>   "In a regular hexagon with side length s = 6, calculate the area of the hexagon.",
>   "A regular dodecagon (12-sided polygon) is inscribed in a circle of radius R = 6. Compute the area of the dodecagon."
>  ],
>  "cyclic_quadrilateral": [
>   "A cyclic quadrilateral has sides of lengths 7, 8, 9, and 10. Calculate the area of the quadrilateral using Brahmagupta's formula."
>  ],
>  "composite_figure": [
>   "A circle is inscribed in a square, and the square has side length 10. Compute the area of the region outside the circle but inside the square."
>  ]
> }
> }
> ```
> - Identifying the core criterion based on inter-sample distinctions is a critical step. Selecting a non-distinctive dimension, such as one previously used by the parent node, leads to a collapse in the subspace partitioning, lowering the diversity of the synthesized data.
>
> Thank you again for your time devoted to our work again! Sincerely, we hope that these clarifications can fully address your concerns, and prompt you to reconsider our scores accordingly! If you have any questions, please do not hesitate to let us know; we welcome any further discussions!
>
> **References**
> [1] Sudalairaj, S., Bhandwaldar, A., Pareja, A., Xu, K., Cox, D. D., & Srivastava, A. (2024). Lab: Large-scale alignment for chatbots. arXiv preprint arXiv:2403.01081.
> [2] Yu, Y., Zhuang, Y., Zhang, J., Meng, Y., Ratner, A. J., Krishna, R., ... & Zhang, C. (2023). Large language model as attributed training data generator: A tale of diversity and bias. Advances in neural information processing systems, 36, 55734-55784.

---

> > ### Comment · Reviewer_4JZM · 2025-08-01
> >
> > Thank you to the authors for their detailed response. All my concerns are resolved. I have updated my score.

---

> > > ### Author Response · Authors · 2025-08-01
> > > **Response to reviewer 4JZM**
> > >
> > > Dear Reviewer 4JZM,
> > >
> > > We sincerely appreciate your thorough feedback and are grateful for your decision to update the score. We welcome any further discussions!
> > >
> > > Sincerely,
> > >
> > > Authors

---

### Official Review · Reviewer_geZ4 · 2025-06-23

**Clarity:** 4
**Significance:** 3
**Originality:** 3
**Rating:** 4
**Confidence:** 3

**Summary:**

This paper introduces TreeSynth: a novel method for synthesis of text data aided by Decision Trees and Large Language Models (LLMs).
TreeSynth leverages an LLM to generate a decision tree to exhaustively enumerate all possible subspaces of the dataset (also called criteria).
This enumeration is then used to enhance diversity and comprehensiveness of the starting dataset, leading to improved downstream ML performance of models trained with such augmented data.

**Questions:**

- Could you provide further numerical insights that demonstrate the increased diversity of data generated by TreeSynth?
- Could you provide some qualitative examples of answers generated by one of the models with TreeSynth data and with another baseline which showcase how the performance improvements are related with the diversity issue tackled by TreeSynth? (For instance, TreeSynth-aided LLMs can answer questions about domains that are underrepresented in the PersonaHub-aided ones)
- Could you provide an informal justification (or a source) in support of the statement made in lines 116-119? Is it the case that the LLM can possibly generate criteria that do not satisfy such a statement? If so, how often does this issue happen? How can it be identified? To what extent does it affect TreeSynth performance? And how can you address it?
- Enumerating all the possible criteria of a dataset can be prohibitive as the size of the decision tree built by TreeSynth grows exponentially in the number of attributes

**Ethical Concerns:**

["NO or VERY MINOR ethics concerns only"]

**Final Justification:**

Having looked at both mine and the other reviewers, I appreciate the authors' answers, which have addressed all my primary concerns except for some minor points.

**Limitations:**

yes

**Paper Formatting Concerns:**

No major issue noticed

**Quality:**

3

**Strengths And Weaknesses:**

## Strengths
+ **The proposed method showcases consistent improvements with respect to the state-of-the-art.** These improvements are consistently measured over several experiments spanning three LLMs and three datasets.
+ **The authors tackled the issue of data diversity in text data.** This is a significant problem in Machine Learning.
+ **Extensive information to reproduce the results.** The authors provided comprehensive information to reproduce the results. Including the prompts used for each experiment.

## Weaknesses
- **Some statements in the main paper are not properly discussed.** For instance, at lines 116-119: “Subsequently, another LLM, proficient in identifying inter-sample distinctions, determines exactly one core criterion δ (e.g., Type of Mathematical Operation). This criterion $\delta$ optimally partitions $X$ into mutually exclusive attribute values $V_X^\delta = {v_j^\delta |j = 1, 2, ..., m}$” lacks a proof or an informal explanation of why this is the case, nor any experimental evidence is shown in support of this statement. A reference should point to such a source if this has been done in the literature.
- **Need for extensive evaluation of TreeSynth and TreeSynth-guided data balancing.** TreeSynth generation quality is only measured by means of data samples’ cosine similarity. Further insights on the data generated by such a method would strengthen the paper’s statements and contributions. Tables that give insights into the generated trees (e.g., how many criteria are generated, depth of the trees, examples of generated criteria) would strengthen the paper as well. TreeSynth-guided data balancing is evaluated in one experiment only (Table 3), in which it is showcased to outperform all other methods. This method could have been analysed more extensively, showcasing the strength of the proposed method.
- **Need for more qualitative analysis.** In papers related to language models, in which quantitative/formal analysis is often infeasible, it is important to provide qualitative analysis of the produced data. The paper does not provide any example of text generated by the TreeSynth-aided models, nor does it provide examples that showcase how improvements of these generated texts are related to the proposed method.

---

> ### Author Response · Authors · 2025-08-01
> **Response to reviewer**
>
> Dear Reviewer geZ4,
>
> We are grateful for your recognition and time! After carefully considering your feedback, we provide the following explanations to address your concerns.
>
> We would like to first briefly highlight the impact of TreeSynth:
> - **Higher diversity and better performance:** Data diversity is crucial for preventing data collapse, avoiding overfitting, and enhancing model performance. With the tree-guided complementarity and exhaustiveness, TreeSynth synthesizes diverse and comprehensive domain-specific data, and demonstrates better performance than human-annotated or other synthetic datasets, **offering a superior solution to data scarcity, collapse, and the high cost of manual annotation** (Sec. 4.2).
> - **Excellent Scalability:** Greater scalability implies that model capabilities can continuously improve with more data, ultimately leading to superior performance. TreeSynth exemplifies this by **effectively scaling datasets and consistently improving downstream performance with increased data volume**, showcasing its significant potential for remarkable model performance (Sec. 4.2).
> - **Effective Re-balance of Existing Datasets:** In addition to synthesizing data from scratch, TreeSynth can also be used to re-balance existing datasets, achieving **improved performance** (Sec. 4.3).
>
> **Question 1 & Weakness 2: Extensive Evaluation of Generation Quality**
> - Beyond cosine similarity, we demonstrate the superior diversity of TreeSynth's synthetic data through two additional analyses: t-SNE visualization and performance on downstream tasks.
>   - The **t-SNE visualizations** in Figures 4 and 7 provide compelling evidence of the data's diversity and comprehensive coverage. They reveal a distribution that is visibly more balanced and diverse than those of human-crafted datasets and peer methods, which directly correspond to the outstanding diversity metrics reported in Tables 1-4. This superiority is attributed to TreeSynth's core methodology: by partitioning the data space from a global viewpoint and synthesizing data within these subspaces, it effectively mitigates model biases and transcends source data limitations.
>   - Given that model performance is highly correlated with data diversity, the **results on downstream tasks serve as practical validation.** TreeSynth not only surpasses baseline methods, but also scales effectively, with performance improving alongside data volume (Figures 5, 6, 8; Tables 5-8). This outcome provides strong evidence of the superiority of its data diversity.
> - As described in Sec. A.5, we empirically set the tree depth to 4. The following table provides statistics on the number of criteria and attribute values at each depth within the spatial partitioning trees that TreeSynth generates for various tasks. We also present the subtree of MATH, with the further partitioning criteria denoted in brackets. The results indicate that as depth increases, the quantities of both criteria and attribute values grow exponentially. This demonstrates that by deepening its structure, TreeSynth continuously partitions the data space and describes the resulting subspaces with more fine-grained attribute values. This capability forms the basis of TreeSynth's superior performance in data diversity, downstream task performance, and scalability.
>
> |**Dataset**|**MetricType**|**RootNode**|**Level1**|**Level2**|**Level3**|**Level4**|**Total**|
> |---|---|---|---|---|---|---|---|
> |**CodeAlpaca**|**NumberofCriteria**|1|12|247|3853|-|4113|
> ||**NumberofAttributeValues**|1|12|247|3853|57683|61796|
> |**GSM8K**|**NumberofCriteria**|1|13|284|3914|-|4212|
> ||**NumberofAttributeValues**|1|13|284|3914|52721|56933|
> |**MATH**|**NumberofCriteria**|1|7|113|1246|-|1367|
> ||**NumberofAttributeValues**|1|7|113|1246|15794|17161|
> ```
> Root(MATH) [Mathematical Domain]
>  |-- Algebra [problem_type]
>   |-- equation_solving [computation_complexity]
>    |-- low [equation_structure]
>     |-- simple_polynomial
>     |-- exponential
>     |-- ...
>    |-- medium [Equation Type]
>     |-- Quadratic
>     |-- Cubic
>     |-- ...
>    |-- ...
>   |-- ...
>  |-- Probability [Problem Complexity]
>   |-- ...
>  |-- ...
> ```
> - Compared to data balancing, **synthesizing data from scratch carries greater weight in our core contribution.** Therefore, we devote more extensive efforts to synthesizing data from scratch, while treating data balancing as an additional feature. Through this supplementary analysis, we demonstrate that TreeSynth-guided data balancing remarkably enhances the distributional balance of existing datasets,  improving data diversity and downstream performance.

---

> ### Author Response · Authors · 2025-08-01
> **Response to reviewer (2/n)**
>
> **Question 2 & Weakness 3: Qualitative Analysis for TreeSynth**
> - To address your concerns, we compare the performance of the Qwen-2.5 7B model on the MATH dataset after training it on MATH-style data generated by either TreeSynth or Persona Hub. As presented in the table below, the model trained with TreeSynth data outperforms the Persona Hub-trained model across all seven sub-domains of MATH. The performance lead is particularly significant in Number Theory, Geometry, and Counting & Probability, with gains of 12.4%, 9.0%, and 8.2%, respectively. This comprehensive and substantial improvement provides empirical evidence for superior diversity and effective data space coverage of the data generated by TreeSynth.
>
> |Domain|PersonaHub|TreeSynth|ImprovementoverPersonaHub|
> |-|-|-|-|
> |NumberTheory|58.7%|71.1%|+12.4%|
> |Geometry|48.9%|57.8%|+9.0%|
> |Counting&Probability|57.0%|65.2%|+8.2%|
> |IntermediateAlgebra|36.4%|44.2%|+7.8%|
> |Prealgebra|79.1%|84.6%|+5.5%|
> |Algebra|83.7%|88.0%|+4.3%|
> |Precalculus|39.2%|42.7%|+3.5%|
> - The following Number Theory case (the content is simplified due to the rebuttal space constraints) demonstrates TreeSynth's superiority over Persona Hub. The TreeSynth-trained LLM correctly computes modular inverses, while Persona fails. This contrast indicates that specialized fields like number theory, which demands precise modular arithmetic computations, may be susceptible to data biases of Persona Hub. TreeSynth's data generation strategy exhibits diverse data and comprehensive coverage, enabling the model to achieve markedly superior performance in specialized mathematical reasoning.
> ---
> **Question**: Compute $(3^{-1}+5^{-1})^{-1} \bmod 31$. Express your answer as an integer from $0$ to $30$, inclusive.
>
> **Correct Answer**: 29
>
> **TreeSynth Solution**: 29
>
> **TreeSynth Reasoning**:
> 1. Find modular inverses: $3^{-1} \equiv 21 \bmod 31$ and $5^{-1} \equiv 25 \bmod 31$
> 2. Add the inverses: $21 + 25 = 46 \equiv 15 \bmod 31$
> 3. Find inverse of 15: $15^{-1} \equiv 29 \bmod 31$ (since $15 \times 29 = 435 \equiv 1 \bmod 31$)
>
> Therefore, $(3^{-1}+5^{-1})^{-1} \equiv 29 \bmod 31$.
>
> **Persona Solution**: 21
>
> **Persona Reasoning**:
> 1. Find modular inverses: $3^{-1} \equiv 21 \bmod 31$ and $5^{-1} \equiv 25 \bmod 31$ ✓
> 2. Add the inverses: $21 + 25 = 46 \equiv 15 \bmod 31$ ✓
> 3. **ERROR**: Claims $15^{-1} \equiv 21 \bmod 31$, but $15 \times 21 = 315 \equiv 5 \bmod 31 \neq 1$ ✗
>
> The final answer is: 21
>
> ---

---

> ### Author Response · Authors · 2025-08-01
> **Response to reviewer (3/n)**
>
> **Question 3 & Weakness 1: Validation of the Criterion Selection Mechanism**
> In the Criterion Determination step, we task an LLM with finding an optimal criterion, $δ$, to partition $l$ pivot samples into $m$ subspaces represented by mutually exclusive attribute values.
> - Initially, we also try to propose theoretical proof of optimality, but find that it is difficult to precisely formulate the data space. We welcome any further discussions here!
> - **This objective is necessary to avoid data collapse.** Assuming that a non-distinctive dimension, such as one previously used by the parent node, is selected,  this will lead to a collapse in the subspace partitioning due to similar attributes of pivot samples, rendering the division ineffective and lowering the diversity of synthesized data.
> - **Prompting the LLM to approximate this objective is feasible with verified effectiveness.** As shown in Figures 17-20, we explicitly instruct the LLM to "Identify exactly ONE core dimension that best distinguishes pivot samples". Given the powerful capabilities of modern LLMs, they generally provide a satisfactory criterion. As shown below, given 10 MATH-style geometry problems, an LLM can identify "geometric_object_type" as the criterion to effectively partition the data.
> ```
> {
> "criterion": "geometric_object_type",
> "attributes": {
>  "triangle": [
>   "A triangle has sides of lengths 7, 24, and 25. Determine the area of the triangle.",
>   "Let ABC be a triangle with AB = 13, BC = 14, and CA = 15. Compute the radius of the circle inscribed in the triangle.",
>   "Let ABC be a triangle where AB = 8, BC = 10, and CA = 12. Find the length of the altitude from A to BC.",
>   "Consider a triangle ABC with AB = 6, BC = 8, and CA = 10. Prove that ABC is a right triangle and then compute the radius of its circumscribed circle.",
>   "In a triangle ABC, the medians from vertices A, B, and C intersect at a point G. If AB = 12, BC = 16, and CA = 20, compute the distance from G to vertex A.",
>   "Let ABC be a triangle where AB = 15, BC = 20, and CA = 25. Determine the coordinates of the orthocenter if vertex A is at (0, 0), B is at (15, 0), and C is at (0, 20)."
>  ],
>  "regular_polygon": [
>   "In a regular hexagon with side length s = 6, calculate the area of the hexagon.",
>   "A regular dodecagon (12-sided polygon) is inscribed in a circle of radius R = 6. Compute the area of the dodecagon."
>  ],
>  "cyclic_quadrilateral": [
>   "A cyclic quadrilateral has sides of lengths 7, 8, 9, and 10. Calculate the area of the quadrilateral using Brahmagupta's formula."
>  ],
>  "composite_figure": [
>   "A circle is inscribed in a square, and the square has side length 10. Compute the area of the region outside the circle but inside the square."
>  ]
> }
> }
> ```
> - **Whether the LLM can possibly generate criteria that do not satisfy such a statement:** Due to the difficulty of formulating the data space, there is no ground truth here to evaluate whether the optimal criteria are generated. However, as mentioned above, this goal is necessary to guide the LLM to produce good partitioning criteria, which is empirically verified to yield good performance. In addition, based on our observations, the strong capabilities of the LLM are sufficient to generate criteria that roughly meet the requirements.
>
> **Question 4: Challenge of Criterion Enumeration**
> - While the total number of nodes and criteria grows exponentially with tree depth, **the complexity of generating a new criterion for any single node increases only linearly.** As illustrated in Figures 17-20, the criterion for a given node is determined by the existing criteria and attribute values of its parent path from root to the current node, as well as its pivot samples. For this simple task, a popular LLM is capable of generating more than 10 layers (i.e., determine the 11th criterion based on 10 previous criteria and 10 pivot samples). Assuming that each node has 10 child nodes, this will result in 10^10 samples, which is generally enough for a specific domain or task.
>
> Thank you for your time! We hope these explanations address your concerns and you can reconsider our scores. If you have any further questions, please do not hesitate to reach out; we welcome any insightful discussions!

---

### Official Review · Reviewer_RdrW · 2025-06-30

**Clarity:** 3
**Significance:** 3
**Originality:** 3
**Rating:** 5
**Confidence:** 3

**Summary:**

The authors present a method for generating synthetic data that is more diverse, inspired by Decision Trees. This method aims to create criteria for subdividing the data space into exclusive regions. To achieve this, they prompt a large language model (LLM) with a data description to generate N diverse samples. Another LLM, proficient in identifying distinctions between samples, determines the core criteria. This process yields a set of instructions used to guide an LLM in generating diverse synthetic data. The authors evaluated their method by starting with popular benchmarking datasets and employing well-known LLMs (both open-source and proprietary) to create synthetic data following the proposed approach. They then fine-tuned smaller models on this generated data, demonstrating performance gains. The baseline for comparison was established using other methods for generating diverse synthetic data. The results indicate that their method outperforms the baseline.

**Questions:**

I also have several questions:

- Is it clear whether the context provided to the LLM is input by the user? The criterion may vary significantly depending on the input.
- What exactly is this "proficient LLM" in identifying inter-sample distinctions that determine the core criteria?
- Evol-Instruct also shows similar diversity in the t-SNE visualizations, and results demonstrate diversity comparable to TreeSynth; however, TreeSynth achieves better outcomes. What explains this discrepancy?

**Ethical Concerns:**

["NO or VERY MINOR ethics concerns only"]

**Final Justification:**

I leaning no maintain my score, as I think these work is a good contribution to the field, despite the dependency of the experiments shown on propertiary LLM. I still feel that adding other source of knowledge will generate better diversity

Overall, a good work, and a worth contribution.

**Limitations:**

The authors stated a set of limitations and the limitations sections.

In my point of view, the main limitations is the dependency of the teacher LLM to generated synthetic data, and I would suggest the authors to better scope the utility of the models to the demostrated use cases as model distillation.

**Paper Formatting Concerns:**

There is no formatting concerns

**Quality:**

3

**Strengths And Weaknesses:**

The paper is well-written and clearly explained, featuring a robust evaluation method and an extensive presentation of results, including effective representations like t-SNE visualizations.

The baseline is effectively presented alongside other similar methods (such as de Evol-Instruct and Persona), and the choice of evaluation method (fine-tuning smaller models) is a strong approach for assessing the performance of the proposed method.

However, I have some concerns regarding the use cases for generating this type of synthetic data. Essentially, this process resembles model distillation, as the method relies entirely on a "teacher" LLM to create diversity within a specific data space. The approach involves a chain of prompts that generate samples and criteria for constructing the Tree, which is then used as guidance to prompt the "teacher" model for generating synthetic data. This dependency is evident in the varying results produced by different base models used in the tests. While the method clearly illustrates model distillation, the introduction may imply that it could generalize and "enhance their specific capabilities."

I recommend that the authors clarify the scope of utility for these models to align with the demonstrated use case of model distillation.

Model distillation indeed offers several advantages, enabling smaller models to be deployed on devices or in fog computing systems, thereby conserving computational resources.

---

> ### Author Rebuttal · Authors · 2025-07-31
>
> Dear Reviewer RdrW,
>
> Thank you very much for your valuable time, dedication, and insightful comments on our work! After reading your feedback carefully, we sincerely prepare the following clarifications after highlighting the significant impact of TreeSynth, and hope that these discussions could fully address your concerns!
>
> First, we will briefly highlight the impact of TreeSynth:
> - **Higher diversity and better performance:** Data diversity is crucial for preventing data collapse, avoiding overfitting, and enhancing model performance. With the tree-guided complementarity and exhaustiveness, TreeSynth synthesizes diverse and comprehensive domain-specific data, and demonstrates better performance than human-annotated or other synthetic datasets, **offering a superior solution to data scarcity, collapse, and the high cost of manual annotation** (Sec. 4.2).
> - **Excellent Scalability:** Greater scalability implies that model capabilities can continuously improve with more data, ultimately leading to superior performance. TreeSynth exemplifies this by **effectively scaling datasets and consistently improving downstream performance with increased data volume**, showcasing its significant potential for remarkable model performance (Sec. 4.2).
> - **Effective Re-balance of Existing Datasets:** In addition to synthesizing data from scratch, TreeSynth can also be used to re-balance existing datasets, achieving **improved performance** (Sec. 4.3).
>
> **Weakness: Clear Paper Scope**
>
> We greatly appreciate your insightful comments! We agree with your observation that our experimental setting resembles that of model distillation, and will clarify this distinction accordingly in our final version. Beyond this similarity, we would like to claim the following points:
>
> - **Our method aligns more closely with data synthesis than model distillation.** Roughly, model distillation focuses on achieving better "efficiency and compression" for a given dataset. In contrast, data synthesis emphasizes how to generate an effective dataset, alleviating data scarcity. The key output of our method is the synthetic dataset itself, instead of a smaller or more efficient "student model", which only serves to validate the effectiveness of synthesized datasets.
> - **This classification is consistent with existing work in the community.** For example, in our baselines, Persona Hub also utilizes GPT-4 to synthesize data and train the Qwen2-7B model, yet it is widely categorized as a data synthesis approach. This provides a strong precedent for defining TreeSynth as a data synthesis method.
>
>
> **Question 1: Whether the Context Provided to the LLM is User-Input**
> - As described in Section 3, the only user input provided to the LLM's prompt is a description of the target dataset. **This description is necessary,** as it instructs the model on what data to synthesize, yet it is simple enough to be easily provided by the users. By requiring only this minimal input, we ensure that TreeSynth remains as concise as possible, achieving **"generating a dataset for a task from a description sentence"**.
> - A better task description is not the core focus of our paper, although it can be optimized by various methods. Specifically, even without deliberately optimized task description, TreeSynth still achieves excellent performance, demonstrating its robustness and effectiveness. The core contribution of this work lies in how to synthesize diverse and comprehensive data through a tree-guided approach. A better task description may lead to further improvements, such as deploying an LLM to optimize the user’s initial input to better guide the data synthesis process. However, this is not the focus of our current study.
>
> **Question 2: Clarification on the Criterion-Determining LLM**
>
> We apologize for the lack of clarity in our previous description!
>
> - Similar to other stages, this "proficient LLM" in Criterion Determination is also an off-the-shelf LLM (e.g., LLaMA-70B, Qwen-72B, or GPT-4o), but is instructed to identify the core criteria with the prompt presented in Figures 17-20. This design leverages the powerful capabilities of existing LLMs, while maintaining TreeSynth's ease of use. The excellent performance of TreeSynth validates the efficacy of this approach.
> - Identifying the core criterion based on inter-sample distinctions is a critical step. Selecting a non-distinctive dimension, such as one previously used by the parent node, leads to a collapse in the subsequent subspace partitioning, rendering the division ineffective and lowering the data diversity.
>
> **Question 3: Why Does TreeSynth Outperform Evol-Instruct Despite Similar Diversity?**
>
> Thanks for your meticulous observation!
>
> - Due to the lack of effective control, **Evol-Instruct often generates deviated data, leading to "spurious" diversity.** Specifically, compared to TreeSynth's tree-guided mechanism that systematically partitions the entire data space into subspaces, the evolutionary process of Evol-Instruct explores outwards from a subspace. This uncontrolled exploration frequently causes synthetic data to deviate significantly from the target data space, exceeding its intended boundaries, which also seems diverse, but degrades downstream task performance.
> - For instance, a GSM8K problem originally solvable by simple division:
>     > "There are 480 zombies in the shopping mall. If the number of zombies doubled every day, how many days ago were there less than 50 zombies in the mall?"
>
>     evolves under Evol-Instruct into a problem requiring complex calculus:
>
>     > "Given that the growth rate of a zombie population in a shopping mall, currently at 480, is proportional to both its current size and the available space within the mall's carrying capacity of 1000, how many days ago was the population less than 50?"
>
>     In contrast, TreeSynth's synthesis process exhibits excellent controllability. It is strictly constrained by the attribute values of each subspace and the data space description from the root node. This design makes TreeSynth demonstrably superior to peer methods in both downstream task performance and scalability.
>
> Thank you again for your constructive comments and your great efforts on our work! Sincerely, we hope that these clarifications can fully address your concerns, and prompt you to reconsider our scores accordingly!

---

> > ### Comment · Reviewer_RdrW · 2025-08-05
> >
> > I appreciate the authors response, and the clarification regarding the scope and the distinction on model distillation.
> >
> > I still feel that relying just on LLM as GPT-4 is a limitation, while other kind of knowledge bases could be used to generare more grounded diversity.

---

> > > ### Author Response · Authors · 2025-08-07
> > > **Further Discussion with Reviewer RdrW**
> > >
> > > Dear Reviewer RdrW,
> > >
> > > Thanks for your insightful follow-up! We fully understand your remaining concerns, and provide the following clarifications!
> > >
> > > - Leveraging knowledge effectively is more critical than adding existing knowledge into models for performance improvement. Even if not perfect, the research community has already explored effective methods for memorizng information. Given that LLMs have been trained on virtually all publicly available internet data, their knowledge base is already broadly aligned with the scope of current human knowledge. The core issue, however, is that the model cannot efficiently apply all the memorized knowledge.
> > > - This limitation has given rise to a new paradigm—self-improvement. Instead of relying on external knowledge acquisition, a model leverages self-generated data to further enhance its capabilities. One of core assumptions of this paradigm is that a model can better utilize its internal knowledge through continued learning (i.e., training), even in the absence of new knowledge inputs. From the perspective of model performance, self-improvement enables continual generalization grounded in the model's existing knowledge.
> > > - Notably, data synthesis is not limited to reproducing known facts for another model to memorize. Rather, it involves strategically guiding the model to utilize its knowledge in diverse ways. The resulting data acts as a medium that exposes downstream models to different forms of knowledge application, ultimately improving task performance. Similar to self-improvement, we hypothesize that data synthesis also holds long-term potential for iterative enhancement: a model synthesizes data that strengthens its ability to apply knowledge; the improved model then generates the next round of data better, forming a loop that progressively boosts generalization on downstream tasks.
> > > - Besides, self-improvement still faces the bottleneck of what kind of data to rely on for progressive gains, and this is where data synthesis will continue to play a central role.
> > >
> > > In summary, we do not view dependence on a knowledge-rich model as a severe limitation. The critical question is how to enable the model to effectively leverage its existing knowledge to improve downstream performance.
> > >
> > > Hope that our clarifications could further address your concerns! Thank you again for this constructive discussion, and we welcome more!
> > >
> > > Best,
> > > Authors

---

### Official Review · Reviewer_W3cd · 2025-07-03

**Clarity:** 3
**Significance:** 3
**Originality:** 3
**Rating:** 5
**Confidence:** 3

**Summary:**

The paper proposes TreeSynth, a method for generating synthetic training data for large language models (LLMs). The core idea is to sample from the intermediate nodes of a tree of LLM generations, which are conditioned on a common root instruction. This tree structure allows TreeSynth to capture both diversity and relevance across multiple levels of reasoning, going beyond traditional sampling techniques like nucleus or top-k sampling. TreeSynth is evaluated by training LLaMA 3.1B and Qwen2.5-7B models on TreeSynth-generated data across four tasks: summarization, closed-book QA, math reasoning, and code generation. The results show that TreeSynth outperforms conventional sampling strategies and even some strong baselines like Self-Instruct or Evol-Instruct in various settings.

**Questions:**

1. Why were only LLaMA 3.1B and Qwen2.5-7B chosen for evaluation? Can TreeSynth generalize to other model architectures or sizes? Experiments on at least one more model family would increase confidence in generality.

2. TreeSynth appears capable of producing a broad range of data types, yet it is only applied to very task-specific settings. Could the authors demonstrate its use on general-purpose instruction tuning data (e.g., FLAN, OIG, or Dolly-style datasets)?

3. What are the computational costs of generating and pruning large generation trees? Could this method scale to hundreds of thousands or millions of examples?

4. Could the relevance scoring function be better calibrated across domains? Are there task-specific biases introduced by using a static scoring model?

5. Is TreeSynth applicable to other stages of LLM development, such as reward model training or RLHF-style pipelines?

**Ethical Concerns:**

["NO or VERY MINOR ethics concerns only"]

**Final Justification:**

The authors' response satisfactorily addressed key concerns. Additional experiments on Gemma-3 models support TreeSynth’s generalizability, and the focus on domain-specific data is well-justified. The cost analysis clarifies the method’s practical feasibility.

While potential applications to RLHF and reward model training intuitively make sense, experimental results would strengthen the paper. General-purpose synthesis and scoring calibration also remain unexplored.

Overall, the core contributions are sound, but some broader claims would benefit from further evidence.

**Limitations:**

yes

**Quality:**

3

**Strengths And Weaknesses:**

## Strengths

- Novel sampling mechanism: The tree-based approach introduces an elegant way to traverse and utilize the latent space of LLM generations, capturing a mix of diverse and high-quality examples by harvesting from both leaf and intermediate nodes.
- Empirical Results: Models trained on TreeSynth data outperform those trained on standard instruction-tuned datasets generated by top-k/top-p sampling or previous bootstrapping techniques.
-Comprehensive evaluation across tasks: The paper evaluates performance on summarization, QA, math reasoning, and coding tasks, providing a good sense of TreeSynth's general applicability within the fine-tuning context.
- Clear methodology: The design of the tree, including beam width, scoring strategies, and node selection policies, is well-explained and easy to follow. Ablation studies on node scoring and position further strengthen the technical argument.


## Weaknesses

- Limited model diversity: The experiments are restricted to only two models: LLaMA 3.1B and Qwen2.5-7B. It remains unclear how TreeSynth generalizes to other model architectures and/or to larger/smaller model sizes. This limits the generality of the claims.
- Narrow scope of data synthesis: The method is only tested on domain- and task-specific data generation (e.g., math reasoning). Despite TreeSynth’s generally task/domain agnostic nature, the paper does not explore its utility for general-purpose instruction tuning or pretraining-style data generation. If the method is capable of broad data synthesis, it should be evaluated on more open-ended tasks or datasets.
- Compute requirements for tree sampling: TreeSynth involves generating a beam of responses at each node recursively, which could be costly at scale. The paper doesn't discuss this overhead or propose methods to make it more efficient.

---

> ### Author Rebuttal · Authors · 2025-07-31
>
> Dear Reviewer W3cd,
>
> We are grateful for your thorough and constructive feedback on our paper! Below, we will try our best to provide detailed responses to each of your comments, and hope that these clarifications will fully relieve your concerns!
>
> First, let's briefly highlight the primary features and impact of TreeSynth.
> - **Higher diversity and better performance:** Data diversity is crucial for preventing data collapse, avoiding overfitting, and enhancing model performance. With the tree-guided complementarity and exhaustiveness, TreeSynth synthesizes diverse and comprehensive domain-specific data, and demonstrates better performance than human-annotated or other synthetic datasets, **offering a superior solution to data scarcity, collapse, and the high cost of manual annotation** (Sec. 4.2).
> - **Excellent Scalability**: Greater scalability implies that model capabilities can continuously improve with more data, ultimately leading to superior performance. TreeSynth exemplifies this by **effectively scaling datasets and consistently improving downstream performance with increased data volume**, showcasing its significant potential for remarkable model performance (Sec. 4.2).
> - **Effective Re-balance of Existing Datasets**: In addition to synthesizing data from scratch, TreeSynth can also be used to re-balance existing datasets, achieving **improved performance** (Sec. 4.3).
>
> **Question 1 & Weakness 1: Broadening TreeSynth Assessment Beyond LLaMA and Qwen**
> - Causes for choosing the LLaMA 3.1B and Qwen 2.5-7B models: Currently, both LLaMA-3.1 8B and Qwen-2.5 7B models are highly representative open-source models, which have been widely accepted in the research community, and are generally sufficient to demonstrate the generalization of TreeSynth. In comparison, baseline methods have narrower scopes. For example, Persona Hub and Evol-Instruct are tested only on Qwen-2 7B and LLaMA 13B models, respectively.
> - Broader scope of model family and sizes: To further showcase the superiority of TreeSynth, we conduct additional experiments on Gemma-3 PT 4B and Gemma-3 PT 12B, which covers both different model series and sizes. As presented in the table below, the results mirror our findings in Sec. 4.2 on LLaMA-3.1 8B and Qwen-2.5 7B: TreeSynth continues to significantly outperform human-annotated data and peer methods, while its downstream performance consistently scales with data volumes. This strongly validates the robustness and generalizability of our method.
>
> | Model         | Benchmark   | Vanilla-0shot | Vanilla-4shot | Code Alpaca | MATH | Temp Sampling | Evol-Instruct | Persona Hub | TreeSynth |
> | :------------ | :---------- | :------------ | :------------ | :---------- | :--- | :------------ | :------------ | :------- | :-------- |
> | **Gemma-3 PT 4B** | **MBPP** | 29.8          | -             | 41.2        | -    | 42.2          | 42.2          | 42.8     | **44.2** |
> |               | **Human Eval**| 33.5          | -             | 34.76       | -    | 35.98         | 37.20         | 37.80    | **42.07** |
> |               | **MATH** | 5.2           | 23.8          | -           | 20.0 | 25.3          | 26.6          | 27.2     | **30.3** |
> | **Gemma-3 PT 12B**| **MBPP** | 54.4          | -             | 56.6        | -    | 57.0          | 56.6          | 56.2     | **57.4** |
> |               | **Human Eval**| 59.1          | -             | 56.10       | -    | 54.27         | 57.32         | 56.71    | **62.80** |
> |               | **MATH** | 23.9          | 31.3          | -           | 34.9 | 47.3          | 49.2          | 50.1     | **54.2** |
>
> | Model | Benchmark | 1k | 2k | 7.5k | 10k | 100k |
> | :--- | :--- | :--- | :--- | :--- | :--- | :--- |
> | **Gemma 3 4b** | **MATH** | 28.5 | - | 30.3 | 30.7 | 35.2 |
> | | **MBPP** | 42.4 | 44.2 | - | 45.4 | 47.8 |
> | | **Human Eval** | 39.02 | 42.07 | - | 43.90 | 45.73 |
> | **Gemma 3 12b**| **MATH** | 53.2 | - | 54.2 | 55.6 | 57.3 |
> | | **MBPP** | 55.2 | 57.4 | - | 58.8 | 60.2 |
> | | **Human Eval**| 61.59 | 62.80 | - | 63.41 | 66.46 |
>
> **Question 2 & Weakness 2: Evaluating TreeSynth for General-Purpose Data Synthesis**
>
> - **The focus of TreeSynth lies on synthesizing data from scratch**. Actually, general-purpose instructions are abundantly collected from the web or human labor, while **domain- and task-specific data is often scarce, necessating the acquirement from effective sources**. Therefore, we focus on synthesizing data for such domains. This addresses a key challenge in LLM customization: the depletion of open-access data and the high cost of human annotation.
> - **Baseline methods**, such as Evol-Instruct and Persona Hub, **also aim to generate domain-specific data**. This also prompts us to focus on  specific domains **for fair comparison**.
> - Furthermore, the task- and domain-agnostic nature of TreeSynth gives it the potential to synthesize general-purpose instruction tuning data simply by adjusting the root node's description to be general-purpose ones. This capability further highlights the generality and significant potential of our method, but synthesizing general-purpose data from scratch seems a bit redundant, given the existing datasets.
>
> **Question 3 & Weakness 3: Computational Costs of TreeSynth**
>
> Thank you for your valuable question! We prepare the following claims for your reference.
>
> - **Compared to efficiency, downstream performance enjoys higher priority.** Specifically, the cost of data generation is a one-time investment, while the resulting downstream models are used continuously. As demonstrated in Figures 5, 6, 8 and Tables 5-8, TreeSynth can synthesize datasets exceeding 100k samples with effectively maintained quality, leading to consistent performance improvements and remarkable scalbility on downstream tasks as data volume increases.
> - **The efficiency of data synthesis should be considered in tandem with downstream training costs.** Specifically, synthetic data requires model training to be converted into performance gains. High-quality data can lead to more significant savings in model training, whose cost is markedly higher than the inference expense for data synthesis. **Therefore, enhancing data quality is a more strategic approach to significantly reduce the greater expense of downstream training.** The result in Figures 5, 6, and 8 substantiate this point: TreeSynth requires only 1k data samples to match or exceed the downstream task performance of a 100k-sample baseline, which precisely highlights its value and efficiency.
> - **TreeSynth can be powered by open-source models, which ensures that its operational cost and overhead are extremely low.** As detailed in the table below, we calculate token consumption when synthesizing 100k-scale datasets of various styles using TreeSynth driven by Llama-3.1-70B-Instruct. Based on the API pricing from DeepInfra for this model ($0.038/$0.12 per million input/output tokens), we estimate that **the cost to synthesize each dataset is less than 10 US dollars.** This finding suggests that in the current era of increasingly affordable computation, the challenge of generating high-quality data becomes more critical than the computational cost itself.
>
> | Dataset | Input Tokens | Output Tokens | Input Cost (USD) | Output Cost (USD) | Total Cost (USD) |
> | :--- | :--- | :--- | :--- | :--- | :--- |
> | **GSM8K Style** | 23.1M | 65.4M | $0.88 | $7.84 | **$8.72** |
> | **MATH Style** | 7.0M | 25.6M | $0.26 | $3.07 | **$3.33** |
> | **Code Alpaca Style** | 25.1M | 56.1M | $0.95 | $6.73 | **$7.68** |
>
> **Question 4: Calibration and Bias of the Relevance Scoring Function**
>
> To ensure we could address your point accurately, we'd like to begin by clarifying a term. When you mention the "relevance scoring function", do you mean the data diversity metric?
>
> - Assuming that is the case, we follow [1], use an embedding model to generate sentence vectors, and calculate their cosine similarity. As a general metric for data diversity, this method is inherently domain-agnostic.
> - Furthermore, a "domain space" is difficult to be defined mathematically, so that the relevance scoring function can be calibrated correspondingly. We haven't found established literature on this topic, and we welcome any further discussions. This should be interesting!
>
> **Question 5: Potential Applications Beyond Instruction Tuning**
> - **The core advantage of TreeSynth lies in synthesizing abundant and diverse datasets with remarkable scalibility from scratch in a task-agnostic manner.** We believe that with further supplemented reward information, TreeSynth-synthesized data can provide significant support for reward model training, RLHF-style pipelines, and other stages of LLM development.
>
> Thank you again for your dedication and constructive feedback to our work! We will update the supplementary experimental results in our paper. Sincerely, we hope that these clarifications can fully address your concerns, and you could reconsider our scores accordingly!
>
> **References**
> [1] Yu, Y., Zhuang, Y., Zhang, J., Meng, Y., Ratner, A. J., Krishna, R., ... & Zhang, C. (2023). Large language model as attributed training data generator: A tale of diversity and bias. Advances in neural information processing systems, 36, 55734-55784.

---

> > ### Author Response · Authors · 2025-08-04
> > **Response to reviewer W3cd**
> >
> > Dear Reviewer W3cd,
> >
> > We fully understand that you may have had a busy schedule recently!
> >
> > Given that the rebuttal deadline is approaching, we genuinely hope that our clarifications have addressed your concerns and will encourage you to reconsider the scores. If you have any further questions, please do not hesitate to discuss them with us！
> >
> > Looking forward to your further feedback!
> >
> > Sincerely,
> >
> > Authors

---

> > > ### Comment · Reviewer_W3cd · 2025-08-05
> > > **Response to authors**
> > >
> > > Thank you for the detailed response. The additional experiments on Gemma-3 models help support the generalizability of TreeSynth across architectures and sizes. I find your arguments for focusing on domain-specific data reasonable, and the cost analysis offers a useful perspective on the practical utility of tree-based generation.
> > >
> > > The clarification regarding TreeSynth’s potential applicability to downstream stages such as reward model training and RLHF is also appreciated, and it strengthens the broader relevance of the method beyond instruction tuning,  though some preliminary experimental results would further strengthen the paper.
> > >
> > > Your response addresses all of my main concerns, and as a result, I have revised my original score to reflect my overall assessment of the paper.

---

### Decision · Program_Chairs · 2025-09-17

**Decision:**

Accept (spotlight)

**Comment:**

This paper uses LLMs to generate diverse synthetic data. The main contributed idea is to automatically partition the data into semantically meaningful disjoint category (e.g., "generate word problems that involve addition and cookies" vs "generate word problems that involve multiplication and pages", where the math operation being addition and the entity being cookies might be decision nodes in the tree), and then sample at each of these nodes to create combinatorial diversity in the synthetic data being generated.

The authors demonstrate strong results across a few datasets, and the reviewers were generally satisfied with additional experiments and clarifications in the rebuttal, with unanimous consensus to clearly accept.

Overall I think the paper is a pretty good LLM pipeline for synthetic data generation in settings where the space of data instances can be semantically partitioned in the way the authors consider -- the result is a sort of "paint by numbers" system somewhere between purely programmatic data generation and fully freeform prompting an LLM for additional data that seems nice.